# Maintenance of Ligament Homeostasis of Spheroid-Colonized Embroidered and Functionalized Scaffolds after 3D Stretch

**DOI:** 10.3390/ijms22158204

**Published:** 2021-07-30

**Authors:** Clemens Gögele, Jens Konrad, Judith Hahn, Annette Breier, Michaela Schröpfer, Michael Meyer, Rudolf Merkel, Bernd Hoffmann, Gundula Schulze-Tanzil

**Affiliations:** 1Institute of Anatomy and Cell Biology, Paracelsus Medical University, Nuremberg and Salzburg, Prof. Ernst Nathan Str. 1, 90419 Nuremberg, Germany; clemens.goegele@pmu.ac.at; 2Department of Biosciences, Paris Lodron University of Salzburg, Hellbrunnerstr. 34, 5020 Salzburg, Austria; 3Institute of Biological Information Processing: IBI-2, Forschungszentrum Jülich, 52425 Jülich, Germany; j.konrad@fz-juelich.de (J.K.); r.merkel@fz-juelich.de (R.M.); b.hoffmann@fz-juelich.de (B.H.); 4Workgroup BioEngineering, Department Materials Engineering, Institute of Polymers Materials, Leibniz-Institut für Polymerforschung Dresden e.V. (IPF), Dresden, Hohe Straße 6, 01069 Dresden, Germany; hahn-judith@ipfdd.de (J.H.); breier@ipfdd.de (A.B.); 5FILK Freiberg Institute GmbH, Meißner Ring 1-5, 09599 Freiberg, Germany; michaela.schroepfer@filkfreiberg.de (M.S.); michael.meyer@filkfreiberg.de (M.M.)

**Keywords:** ligament tissue engineering, spheroids, ACL, 3D stretch, embroidered functionalized scaffold, polylactic acid (PLA), fluorine, poly(l-lactide-co-ε-caprolactone (P(LA-CL))

## Abstract

Anterior cruciate ligament (ACL) ruptures are usually treated with autograft implantation to prevent knee instability. Tissue engineered ACL reconstruction is becoming promising to circumvent autograft limitations. The aim was to evaluate the influence of cyclic stretch on lapine (L) ACL fibroblasts on embroidered scaffolds with respect to adhesion, DNA and sulphated glycosaminoglycan (sGAG) contents, gene expression of ligament-associated extracellular matrix genes, such as type I collagen, decorin, tenascin C, tenomodulin, gap junctional connexin 43 and the transcription factor Mohawk. Control scaffolds and those functionalized by gas phase fluorination and cross-linked collagen foam were either pre-cultured with a suspension or with spheroids of LACL cells before being subjected to cyclic stretch (4%, 0.11 Hz, 3 days). Stretch increased significantly the scaffold area colonized with cells but impaired sGAGs and decorin gene expression (functionalized scaffolds seeded with cell suspension). Stretching increased tenascin C, connexin 43 and Mohawk but decreased decorin gene expression (control scaffolds seeded with cell suspension). Pre-cultivation of functionalized scaffolds with spheroids might be the more suitable method for maintaining ligamentogenesis in 3D scaffolds compared to using a cell suspension due to a significantly higher sGAG content in response to stretching and type I collagen gene expression in functionalized scaffolds.

## 1. Introduction

The anterior cruciate ligament (ACL) is an intraarticular ligament that stabilizes the knee joint and is one of the most frequently injured structures of the knee [1]. It is characterized by a low cell density and a high amount of extracellular matrix (ECM) [2]. The specialized, spindle shaped ACL fibroblasts have a lower proliferation potential compared to cells derived from other ligaments [3]. Due to limited blood supply the self-healing abilities of the ACL are strongly impaired [1]. Furthermore, a ruptured ACL can not only cause knee pain and knee instability but is also often associated with meniscus injuries and leads in the longterm to osteoarthritis. Consequently reconstructions are frequently performed [4,5,6,7]. However, the current gold standard, using autologous tissues such as parts of the patellar ligament with its bony patellar and tibial attachments or the hamstring tendons does not always withstand the mechanical stress and also leads to the so-called “donor site morbidity” [8]. For this reason, the use of a suitable biomaterial-based transplant has been considered for a long time [9,10]. This requires scaffolds that allow the cell adhesion and promote the production of ECM to build up a new ACL [11]. Biomaterials that can be used as such scaffolds have to fulfil certain requirements. Cyto- and biocompatibility, mechanical stress resiliency and slow degradation in vivo are prerequisites for ACL tissue engineering [12]. It has already been shown that the combination of poly(l-lactide-co-ε-caprolactone) (P(LA-CL))/polylactic acid (PLA) is suitable for ligament tissue engineering, not only because of their cytocompatibility, but also in view of their processability [13,14,15]. Three-dimensional (3D) scaffolds, consisting of P(LA-CL)/PLA threads can be produced by the embroidery technology [13,14,15,16,17,18]. Despite the successful colonization of the embroidered P(LA-CL)/PLA scaffolds with LACL derived fibroblasts, a functionalization of the scaffolds by fluorination and crosslinked collagen foam had to be performed to achieve an additional improvement of cell adhesion [15,19]. As collagen, especially type I collagen, is the main component of the ECM, it gives the tissue the necessary tension-stability [1,20]. Beside chemical cues, mechanostimulation is crucial for the development of ligaments and tendons [21,22,23]. From this it can be concluded that appropriate mechanostimulation has a significant inductive influence on ligament derived fibroblasts with regard to the production and maturation of the ECM. It has been shown in vitro that mechanostimulation has a stimulatory effect on the healing process of ligament and tendon tissue [24,25,26,27]. As a prerequisite for a scaffold suitable for ACL reconstruction, withstanding tensile load must be guaranteed. Besides the required viscoelasticity of the material, it is also important to ensure cell survival. It has already been shown that the mechanical stimuli applied to cells not only facilitate their proliferation and differentiation but also their ECM gene expression [28,29]. The tenogenic differentiation demonstrated by expression of the tendon-related transcription factor scleraxis and expression of ligament associated ECM components, such as types I and III collagens, tenascin C as well as fibronectin could be induced in human mesenchymal stem cells by cyclic stretching [30,31,32,33]. However, it is not yet clear which parameters of stretch (amplitude, frequency, direction, duration) offer an optimal condition for teno- and ligamentogenic differentiation. Ligament derived fibroblasts exposed to 10% stretch under 2D and to 5% stretch (at 1 Hz) under 3D conditions responded with a higher type I collagen expression [26,31]. It was reported that in vivo ACL length and knee flexion are inversely related and that the native ACL can withstand a maximum elongation of 5% before rupturing [34,35].

In addition to the unresolved optimal selection of stretch parameters, it also remains to be explored which cell colonization strategy for scaffolds is most suitable for ligament tissue engineering. In addition to using cell suspensions for scaffold seeding, the trend goes more and more towards 3D pre-culturing with the idea to facilitate redifferentiation to re-induce the differentiated cell phenotype and achieve directed seeding. Spherical 3D cell clusters (spheroids) represent an alternative scaffold colonization approach by allowing zonal and directional colonization [36]. Most importantly, dedifferentiation of tenocytes [37] or ligamentocytes in 2D culture can be avoided if they are cultivated in 3D high cell density culture [38]. Consequently, the 3D spheroid culture establishes a much closer relationship to the native tissue than the 2D culture and also promotes stable ligament associated marker expression [17,39,40].

The aim of the present study was to ensure the ligament-derived fibroblast adhesion and maintenance of their ligament phenotype in 3D stretched ACL scaffolds. The hypothesis of this study is that stretch facilitates LACL fibroblasts’ adhesion, proliferation and ECM production. A second assumption was that a controlled colonization of the scaffolds with spheroids is promoted by cyclic stretch.

## 2. Results

### 2.1. Cyclic Stretch Induced a Homogenous Cell Distribution

Functional scaffolds for tissue engineering should by cytocompatible and allow homogenous cell adhesion (Figure 1). For this reason, vitality assays were performed to distinguish between vital and dead cells. Only a few dead cells could be detected on the functionalized and non-functionalized scaffolds, irrespectively of the seeding strategy applied (Figure 2A–I). By using the cell suspension method for scaffold seeding, a high number of green and hence, vital LACL derived fibroblasts was observed widespread on both, non-functionalized and functionalized scaffolds (Figure 2A–D). The functionalization led to a highly significant larger surface area colonized with cells than the non-functionalized scaffolds (29.5% ± 13.5 versus 10.4% ± 4.9, cell suspension method).

Unstimulated scaffolds showed fewer cells in comparison to the stimulated scaffold, where the LACL derived fibroblasts were also more homogenously distributed on both scaffold types (Figure 2A–I). Especially, the functionalized and stretched scaffold showed a high amount of vital and elongated cells on the single threads. Accordingly, the calculation of the colonized scaffold surface showed that mechanical stimulation was associated with a larger surface area of the scaffold covered with LACL derived cells in comparison to the unstimulated ones, not only in the non-functionalized, but also in the functionalized scaffolds (Figure 3A). This difference was highly significant for the functionalized scaffolds seeded with a cell suspension.

By using the spheroid method, the spheroids were more intimately integrated between the threads of stretched non-functionalized scaffolds in comparison to the unstimulated ones, where fewer cells were emigrating out of the spheroid (Figure 2F–I). The mean size of the spheroids (50,000 cells, 2 days of spheroid assembly) was 613.45 ± 175.96 µm before placed on the scaffold and 593.57 ± 130.71 µm after the entire culturing period of 8 days on the scaffolds.

### 2.2. Migration into Inner Parts of the Scaffold

The colonized scaffold surface was significantly larger in stimulated functionalized scaffolds pre-cultivated with the cell suspension in comparison to the unstimulated functionalized scaffolds, but also compared to the stimulated control scaffolds (Figure 3A). There was a highly significant reduction of cell migration into inner parts of the stretched non-functionalized scaffolds in comparison to the unstimulated control scaffolds (395.1 µm ± 161.6 versus 620.2 µm ± 147.3) in regard to the suspension pre-culture (Figure 3B). However, the difference between the stimulated and unstimulated functionalized scaffolds seeded using a cell suspension did not reach the significance level (561.2 µm ± 207.8 versus 510.4 µm ± 187.2) (Figure 3B). Despite not reaching the significance level, mechanostimulation was also associated with amplified cell spreading compared to the non-stimulated controls in the functionalized scaffolds pre-cultured with spheroids (19.1% ± 13.6 versus 16.9% ± 14.1) (Figure 3C). In contrast, the stimulated non-functionalized scaffolds with the spheroid pre-culture had a highly significant higher migration depth of LACL derived fibroblasts than the stimulated functionalized scaffolds (572.9 µm ± 144.6 versus 369.0 µm ± 126.1) (Figure 3D).

### 2.3. Cyclic Strain Influenced Proliferation and Cytoskeletal Protein Expression

Based on the identified differences we next analyzed the expression levels of the proliferation marker Ki67 and alpha smooth muscle actin (αSMA), a myofibroblast marker after preincubation with the suspension culture.

Ki67 is an intranuclear proliferation marker. Using the same seeding technique a higher number of Ki67 positive cell nuclei could be observed in the functionalized compared to the control scaffolds (Figure 4A–D, for the overlay of immunocytochemical staining please refer to Appendix A). Immunocytochemical staining of αSMA was performed to show the fibroblast-to-myofibroblast transition and the possible inductive effect of stretching on this process. After a pre-cultivation of the scaffolds with the cell suspension the expression of αSMA was higher in stimulated (control and functionalized) scaffolds in comparison to the unstimulated ones (Figure 4E–H, for the overlay of immunocytochemical staining please refer to Appendix A).

To achieve semiquantitative results for both targets, Ki67 and αSMA immunoreactivity was calculated using ImageJ. Despite the differences did not reach the significance level for both, in response to mechanostimulation more cell nuclei with immunoreactivity for Ki67 and more cells expressing αSMA could be detected in comparison to the unstimulated scaffolds seeded with a cell suspension (Figure 4I,J).

Interestingly, cells on functionalized scaffolds that were seeded with spheroids showed a stronger Ki67 expression in comparison to the control scaffolds which displayed only few Ki67 immunoreactive cell nuclei (Figure 5A–D). Especially the functionalized scaffolds showed a lot of Ki67 expression inside the spheroids, but also in emigrating cells (Figure 5A–D, for the overlay of immunocytochemical staining please refer to Appendix A). The αSMA protein was expressed not only inside the spheroids, but also by the emigrating cells on the control and also on the functionalized scaffolds (Figure 5E–H, for the overlay of immunocytochemical staining please refer to Appendix A). To achieve semiquantitative results for both targets, Ki67 and αSMA immunoreactivity was calculated using ImageJ. Although the differences in immunoreactivity did not reach the significance level in both cases, the calculation of the number of Ki67 immunoreactive cells indicated that cyclic stretch led to a higher number of Ki67 positive cell and a higher expression of αSMA compared to non-stretched scaffolds seeded with spheroids (Figure 5I,J).

### 2.4. Cell Morphology Shown by Histological Staining Using Sirius Red

Sirius red staining was used to examine the distribution of type I and III collagens in unstimulated and cyclically stretched scaffolds as well as the native LACL for comparison. For all samples, type I collagen was predominant (red-orange), with only a few visible type III collagen fibers (yellow). In the control scaffolds type I collagen could be detected in the suspension as well as in the scaffolds seeded with the spheroid culture. Stretched fibroblast seemed to have a higher type I collagen production on the single threads in comparison to the unstimulated ones. The collagen foam belonging to the functionalization was also red-orange colored. Nevertheless, single cells and spheroids with their emigrating cells could be observed surrounded by pericellular red type I collagen staining (Figure 6).

### 2.5. Enhanced Cell Amount and sGAG Content by Strain

To further verify Ki67 results, the DNA content was measured after 72 h of stimulation to estimate the cell amount per cubic centimeter of scaffold volume. The native LACLs contained the highest amount of DNA in comparison to the tissue engineered constructs. However, only the difference between native LACL and functionalized scaffolds seeded with spheroids (unstimulated: 50.26 ± 14.05; stimulated: 67.40 ± 48.33 DNA amount (pg) per cm^3^) was significant (Figure 7A).

In general, the DNA content was higher in non-functionalized than in functionalized scaffolds for the suspension pre-culturing method. Furthermore, mechanostimulation increased the DNA content of LACL derived fibroblasts in the functionalized scaffolds (not significant) (Figure 7A).

Despite not reaching the significance level, stretch induced an upregulation of the sulphated glycosaminoglycan (sGAG) content per cell in the non-functionalized scaffolds with suspension pre-culture, but also in the functionalized scaffolds with spheroid pre-culturing. In the control scaffolds seeded with spheroids, cyclic stretch led to a not significant suppression whereas in the functionalized scaffolds colonized with a cell suspension, cyclic stretch led to a significant suppression of the sGAG content. In the absence of stretch functionalized scaffolds seeded with a cell suspension displayed a significantly higher sGAG content compared to the unstimulated control scaffolds. The sGAG content of the native LACLs (36.8 ± 12.9 pg/cell) was significantly higher in comparison to the stimulated control scaffolds with spheroid pre-culturing (19.2 ± 1.1 pg/cell). Interestingly, the sGAG content of stimulated functionalized scaffolds with spheroids (46.9 ± 7.7) was significantly higher than that of stimulated functionalized scaffolds with cell suspension pre-culture (27.8 ± 0.8 pg/cell) and also significantly higher than that of stimulated control scaffolds with spheroid pre-culture (19.2 ± 1.1 pg/cell) (Figure 7B).

### 2.6. Relative Gene Expression of Ligament Associated Genes and Connexin 43

Gene expression of the main ligament-associated ECM components such as type I collagen, decorin, tenascin C, the ligament-related glycoprotein tenomodulin, the transcription factor Mohawk and connexin 43 were evaluated in comparison to native LACL tissue.

In comparison to the native LACL type I collagen gene expression was generally significantly lower in cells cultured on all stretched and non-stretched scaffolds except for those on the functionalized scaffolds seeded with spheroids. The latter, stretched and non-stretched cultures, showed a significantly higher type I collagen expression than the native LACL. In addition, the relative gene expression levels of type I collagen were in general higher in cells on the functionalized scaffolds in comparison to those on the non-functionalized scaffolds. Furthermore, the spheroid pre-culturing method evoked a higher relative gene expression level of type I collagen than the suspension pre-culturing method. Nevertheless, mechanostimulation had no significant effect on type I collagen expression of cells in the investigated setting (Figure 8A).

The relative decorin gene expression of the native LACL was generally significantly higher compared to all investigated scaffold cultures.

It was also significantly higher in the absence of stretching in cells on the suspension pre-cultured control and functionalized scaffolds in comparison to those on the same scaffolds seeded with the spheroid pre-culturing method. The mechanostimulation induced a significant reduction of the relative gene expression of decorin in cells on control and functionalized scaffolds pre-cultured with cell suspension but not in functionalized scaffolds seeded with spheroids. Cells on functionalized scaffolds seeded with spheroids, irrespectively whether stimulated or not, exerted a significantly higher decorin gene expression compared to the control scaffolds (Figure 8B).

Cells on scaffolds seeded with spheroids (stimulated or not) and on mechanostimulated scaffolds seeded with the cell suspension expressed significantly more tenascin C mRNA than cells in the native LACL. Stimulation induced a significant upregulation of the relative gene expression of tenascin C in the suspension pre-cultured control and functionalized scaffolds. Cells on the functionalized mechanostimulated scaffolds seeded with spheroids expressed significantly more tenascin C than control scaffolds seeded with spheroids (Figure 8C).

The relative gene expression of the ligament-related glycoprotein tenomodulin in the native LACL revealed no significant difference compared to all investigated scaffolds, irrespectively whether stimulated or not. It was upregulated in the control scaffolds through stretch, not only in the suspension but also in the spheroid pre-cultured scaffolds (not significant). Cyclic stretch reduced the relative gene expression of tenomodulin in the functionalized scaffolds with both colonization methods (not significant) (Figure 8D).

There was a significantly higher expression of the transcription factor Mohawk in the native LACL in comparison to the cells cultured on all scaffold variants. The cyclic stretch only induced a significant upregulation of the relative gene expression of Mohawk in cells seeded on the control scaffolds with the suspension pre-culturing method compared to cells on the same scaffolds either non-stretched but also compared to the stretched functionalized scaffolds, both seeded in a similar manner or stretched control scaffolds seeded with spheroids (Figure 8E).

Compared to the native LACL in all control scaffold cultures a significant upregulation of connexin 43 could be observed. Mechanostimulation induced a significant upregulation of the relative gene expression of connexin 43 in the control scaffold pre-colonized with cell suspension. The upregulation of the relative gene expression of connexin 43 in the stimulated suspension pre-cultured control scaffolds was significantly higher compared to that of the cells on stretched functionalized scaffolds seeded in a similar manner or to mechanostimulated control scaffolds seeded with spheroids. Non-functionalized scaffolds seeded with spheroids expressed more connexin 43 than functionalized scaffolds with similar cell seeding, irrespectively, whether stretched or not (Figure 8F).

## 3. Discussion

ACL reconstructions, especially using autografts represent a significant improvement in the quality of life by restoring knee joint stability. In view of limitations in autograft availability tissue engineered ACL could present a future option [41]. The ACL is exposed to repetitive tension [1] and also ligament healing benefits from mechanostimulation [27]. Moreover, the tensile load plays a major role in the homeostasis of tendons and ligaments, especially the ACL [28,42,43]. It is also well known that the typically spindle-shaped ACL fibroblast morphology developed under the influence of cyclic stretch [42,44]. However, the influence of cyclic stretch on cellular processes during ligament tissue engineering have to be better understood and it could be hypothesized that the results of ACL tissue engineering might be improved by applying appropriate cyclic tension in vitro. Hence, the proposed approach is a potential strategy to address this topic.

Therefore, the aim of the present study was to investigate the effect of cyclic stretch on LACL derived fibroblasts’ vitality, migration and ligament associated ECM expression on embroidered P(LA-CL)/PLA scaffolds. The scaffolds used have been demonstrated in previous studies to be biomechanically stable enough [45] and highly cytocompatible [15]. Vitality assay suggested, that cyclic stretch enhanced the cell adhesion and spreading of the fibroblasts in the present study. The colonized area was larger in functionalized scaffolds reflecting a higher number of adhering cells in comparison to the control scaffold as the collagen foam might allow a superior cell adhesion. A high cell migration could also contribute to the size of colonized surface of the scaffolds. Another study using uniaxially stretched silicone membranes seeded with tenocytes indicated a stimulatory effect of stretching on cell migration [46]. For this reason, the migration depth into inner parts of the scaffolds was measured in the present study. It was greater in the control scaffolds in comparison to the functionalized scaffolds, probably due to the fact, that these scaffolds contained no collagen foam, which might inhibit to some degree the cell migration into deeper scaffold areas. The collagen foam covers the scaffold surface and is distributed between the threads, thereby the cells have to penetrate the foam to get into inner parts of the scaffold. The migration depth could not be directly compared between both pre-culturing approaches due to differences in the pre-cultivation time. Spheroids required more time (2 days of spheroid formation) and a static culturing approach for 5 days to guarantee firm adherence to allow stretching. Furthermore, one has to consider that suspension pre-culturing in a dynamical device allows cell migration from all sides into the scaffold center whereas spheroids were placed only on the surface. Hence, differences of the migration depth between the suspension and spheroid pre-cultivation method could be explained by these differences. Nevertheless, surprising was the observation that cyclic stretching of the control scaffold seeded with a cell suspension significantly decreased the migration depth. The question arises whether the pores without collagen foam supplementation somehow collapse during stretching becoming too small for cell penetration whereas in the functionalized scaffolds the foam prevents this. However, it became only evident in suspension pre-cultured scaffolds. In the spheroid-based approach the migration depth was generally around one third lower. This parameter might also depend on the spheroid size used. Rather large spheroids (50,000 cells per spheroid with 593 ± 13 µm diameter) were used in this study to allow a defined positioning and prevent falling through the pores of the scaffold. This bears the risk of impaired nutrition of the core of the spheroid as around 250 µm can be nourished by diffusion [47]. The spheroids used in the present study could be at the nutrition limit by diffusion. As ligaments are known as a bradytrophic tissue with poor blood supply these hypoxical and malnutrition conditions might not affect cell viability. Nevertheless, when the spheroids are applied to the scaffolds they become ellipsoid during adhesion leading to a smaller diffusion distance.

Cell proliferation could also be a reason leading to higher colonized areas on the scaffolds. In contrast to other authors, who had reported that mechanical load (10%, 1 Hz, 120 min, using the 2D flexcell system) increased the proliferation of human tenocytes by upregulating substance P [48] our results based on a protocol of only 4% stretch at 0.11 Hz for 2 days under 3D conditions resulted in no significant differences. To analyze this, we immunolabeled the Ki67 antigen and measured the DNA content. It is possible that the LACL derived cells suffered from overload in this study with 4% stretch, although we had shown previously that the same cells showed a significantly higher proliferation rate under 2D conditions, but also when emigrating from 3D spheroids with 14% stretch [49]. However, it remains debatable to directly compare 3D with 2D stretch conditions since the cells have to sustain the 3D deformation of their complete environment under 3D conditions. In the situation of 2D strain, they can to some degree avoid the stress arising from stretching of their underlayment by cell reorientation in zero stretch direction. Ki67 protein is active during all active phases of cell cycle (G1, S, G2 and mitosis) with the exception of the G0 stage [50]. Despite the effect was not significant, our immunocytochemical staining suggested that LACL derived fibroblasts might proliferate more rapidly on functionalized scaffolds, particularly, when exposed to stretch, in comparison to the control scaffold. As the DNA content did not significantly differ between the scaffold variants, the question arises whether stretching might be associated with some cell loss which hides the effects of proliferation. To prove this, the DNA contents before and after stretching have to be compared in future.

The active myofibroblastic phenotype is detectable in native ACLs and ACL derived fibroblasts by expressing αSMA [18,40]. Due to its contractile properties and the synthesis of ECM, especially type I collagen, myofibroblasts are essential for wound closure and tissue repair [51,52,53]. Results of immunocytochemical staining showed substantial αSMA expression in stretched scaffolds, particularly in the suspension pre-colonized scaffolds but also in the spheroid pre-colonized scaffolds. This finding might indicate the flexibility of LACL derived fibroblasts, capable for transition to a phenotype characterized by contractile properties induced by 4% cyclic stretch which could be associated with enhanced regenerative potential but also with the risk of scarring. We could hypothesize that increased expression of αSMA might occur particularly in the most stretched single filaments of the scaffold and thus, a fibroblast to myofibroblast transition took place [54,55], probably in response to the microenvironmental conditions.

Granato et al. (2017) showed that the expression of αSMA was downregulated after 72 h in spheroid cultures, which indicated a myofibroblast deactivation [56]. Stretching led obviously to a myofibroblast’s transition as indicated by enhanced αSMA. Not only the activation and proliferation of myofibroblasts are essential for scaffold colonization but also their survival and migration play an important role in ligament tissue engineering and were triggered through the focal adhesion kinase (FAK) [57,58]. Components of cellular focal adhesions such as vinculin, talin and paxillin interact with this kinase and are involved as parts of the focal adhesion complexes in the cell-ECM binding through integrins which can also be shown in spheroids used as tumor model [59]. The presence of multiple focal adhesions could be demonstrated by F-actin staining. Formation of F-actin was clearly detectable. Accordingly, as an adaption to stretching, a dynamic remodeling process of the actin cytoskeleton was detectable showing more pronounced and orientated stress fibers, mostly running parallel to the fibers. As the cells are still mobile and not sessile, migrating on the fibers into the scaffolds, their focal adhesions appear rather small. A similar effect has been reported for tracheal smooth muscle cells and mouse embryo fibroblasts [60].

To check the maintenance of the ligamentocyte-specific phenotype in embroidered scaffolds, expression of several tendon/ligament related genes, such as type I collagen, decorin, tenascin C, tenomodulin and Mohawk was investigated in this work. High cell density 3D cultures inhibit the dedifferentiation process of tendon fibroblasts observed in 2D monolayer culture [37,38,39,61]. Due to the fact that ECM synthesis is upregulated in 3D culture systems (such as spheroids, pellets, high density cultures) [38,39], we hypothesize that additional cyclic stretch promotes and enhances ECM formation.

Type I collagen, as a major component of tendons and ligaments is responsible for the mechanical strength of the tissue [1,62]. Hence, it is an important observation that type I collagen expression did not significantly differ between the native LACL and all investigated scaffold cultures. Not only our results but also those of other authors showed that a spheroid pre-cultivation could help to amplify type I collagen gene expression in comparison to a 2D culture [63] or compared with suspension culture as observed in the present study. A typical phenomenon, which had been reported also by another research group was that spheroids fused very easily on matrices [64] and hence, with the scaffold. This is a sign of tissue formation, the colonized area on our scaffolds had shown this and therefore, this 3D colonization strategy is a powerful tool for ligament tissue engineering. Against our hypothesis, the relative gene expression of type I collagen in stretched scaffolds was downregulated in comparison to unstimulated samples. This phenomenon that stretch led to a downregulation of type I collagen expression in 3D polymer scaffolds could also be observed by another research group, treating patellar tendon fibroblasts in a chitosan-hyaluronan hybrid matrix with 5% stretch, 0.5 Hz for 14 days (18 h stretch and 6 h rest per day) [65]. Additionally, 2% strain at a frequency of 1 Hz with different stimulation durations (4, 8, 24 h) had a suppressive effect on the relative type I collagen gene expression of equine adipose derived MSCs in tendon scaffolds [66]. Kreja et al. reported an upregulation of type I collagen at 2% and 5% strain in polymer scaffolds seeded with MSCs or ligamentocytes after a short stimulation period of only 1 h stretch per day [31] and also 3% strain (0.2 Hz) evoked an increase in type I collagen after 7 days in MSCs [67]. It could be possible that not only the stretch parameters but also the stretch duration plays a decisive role in the type I collagen expression and at least also the cell carrier matrix. The degree of stretching was 4% in the present study and hence, in the range used in the above cited publications and lower as in other studies [68]. Tendon slices exposed at least up to 5% strain do not show damage [69]. The risk of irreversible plastic deformation of the polymer scaffold can be excluded as the embroidering pattern provides by entangling the threads in response to stretching a reserve representing the so called toe region in the stress-strain curve of the scaffold [14,45]. A rather low frequency of 0.11 Hz was used which is helpful to minimize artefacts produced by fluid flow.

Therefore, it could be a time-dependent problem, as compared with another study, which showed that the relative type I collagen expression of human embryonic stem cells increased with a longer stretch duration on 3D scaffolds [70]. It has already been shown that the relative gene expression of type I collagen is higher in a 3D environment than in monolayer 2D culture [39]. Another possible influence factor could be the addition of 10% FBS, because it has been shown in cultured cardiac fibroblasts that the higher the FBS concentration, the lower is the type I collagen expression [71]. However, it is interesting to see, that in contrast to another study where growth supplements were required to maintain ECM expression in equine tenocyte spheroids, our LACL derived fibroblast did not lose their ability for expression of the type I collagen during a cultivation time of 5 days and an additional stretching time of 3 days without further growth supplementation [39]. However, it is unclear why 4% stretching did not lead to an increase in the gene expression of type I collagen in the spheroid culture. This might depend on a possibly spheroid-specific response to stretch such as improved availability of FBS in the core of the spheroids due to elevated fluid flow. Nevertheless, protein deposition of collagen could be shown in all cultures by Sirius red staining.

In addition to type I collagen, the small leucine-rich proteoglycan, decorin, regulates the collagen fibrillogenesis being a fundamental ligament ECM protein [72]. In comparison to other studies, which showed an upregulation of decorin in response to up to 10% strain [73,74] decorin expression was generally lower in the scaffolds than in the native LACL, except for the unstimulated functionalized scaffolds seeded with a cell-suspension. Possibly parts of the enthesis which contains fibrocartilage rich in proteoglycan expression were included in the analysis. Stretching impaired it significantly in the same culturing approach. This effect on decorin was in agreement with the measured sGAG release and could be explained by the fact that decorin is the most common proteoglycan in tendon and ligaments presenting probably a majority of sGAG side chains in both tissues. Proteoglycans might be rather regulated by pressure which can arise from fiber twisting but it does not occur in the uniaxial stretching device used in this study.

The ECM glycoprotein, tenascin C is expressed during embryogenesis in the myotendinous junction and is therefore, a marker for teno-/ligamentogenesis [75]. It was regulated by mechanical loading in the rat Achilles tendon [75]. The significant upregulation of the relative gene expression of tenascin C through cyclic stretch in the cell-suspension pre-culturing approach on both scaffolds and a generally significantly higher expression in comparison to the native LACL suggest the maintenance of the ligament phenotype.

Tenomodulin, is a member of the type II transmembrane glycoproteins and a late phenotypic marker of tendon and ligament formation [76]. It is also known that a loss of tenomodulin reduced the proliferation of tendon stem/progenitor cells and also the adhesion to type I collagen [77]. Generally, no significant difference to the native LACL could be detected in all cultured scaffolds. It has been reported that the relative gene expression of tenomodulin depends on the stretch intensity and the ECM [78,79].

Mohawk is a ligament-associated transcription factor for ligaments, which plays an important role during tendon development and maintenance [80] and is also involved in ECM regulation by mechanical stimuli [81]. A knock down of Mohawk expression by shRNA led to a downregulation of type I collagen, type XIV collagen and fibromodulin in bone marrow-derived stem cells and tenocytes in the study of Yang et al. [82]. Only the non-functionalized scaffolds seeded with a LACL cell suspension showed a significant increase in Mohawk in response to stretching. Due to the above mentioned interrelation between Mohawk and type I collagen gene expression this might be related to the low type I collagen expression in these samples.

Mechanical signals were transduced through connexin 43 in ligament derived fibroblasts [83] and tenocytes [84,85]. An upregulation of the relative gene expression of connexin 43 could not only be proven in periodontal ligament cells [86] but also in our LACLs cultured on control scaffolds. The general opinion is that the physiological application of 4% strain might increase the intercellular communication [87,88]. However, the relative gene expression of connexin 43 was only upregulated in the suspension pre-cultured control scaffolds through stretch but remained lower on functionalized scaffolds. It was shown that a high concentration of sodium fluoride impaired the intercellular communication via gap junctions in rat osteoblasts [89]. Hence, the effect of fluorination has to be considered. As only the gene expression was detected here one has to consider that densely packed cells in spheroids exert already a high protein expression of connexin 43 which is impaired by a drop in gene expression to allow cell emigration onto the scaffold fibers. Nevertheless, the connexin 43 expression was except for the functionalized scaffolds higher than in the native LACL. The reestablishment of a communicating cell network in the scaffold cultures might require a transient connexin 43 upregulation.

## 4. Materials and Methods

The experimental setting of the two different colonization methods and the two different scaffolds is shown in a schematic overview (Figure 1).

### 4.1. Preparation of the Embroidered P(LA-CL)/PLA Scaffolds

For the embroidering (JCZ 0209-550, ZSK Stickmaschinen GmbH, Krefeld, Germany) a monofilament suture thread made of P(LA-CL) (USP 7-0, Gunze Ltd., Osaka, Japan) and a melt spun multifilament consisting of six filaments made of PLA (Tt = 155 dtex, Ingeo biopolymer 6202D, NatureWorks, Minnetonka, MN, USA), melt and spun at IPF (Dresden, Germany) were used. Scaffolds were composed of three plies with a zig-zag pattern design (1.8 mm stitch length, 15° stitch angle and 0.2 mm duplication shift). In an additional manufacturing step, these three plies were stacked and locked together to get a 3D scaffold. The scaffold had a dimension of 30 mm length, a width of 4 mm and a thickness of 1 mm (Figure 1F,G). The pores were between 100–200 µm and the whole scaffold had a porosity of more than 65–75% [90]. The embroidering process was described previously [15,45]. The control scaffolds with no functionalization were then directly used after sterilization in the cell culture.

### 4.2. Functionalization of the Scaffolds

Functionalization of the scaffolds was performed to achieve higher cell attachment rates and increase cell activity [19,91]. Moreover, the gas fluorination applied increased also the adhesion of the collagen foam at the scaffold threads (unpublished results). The functionalization of the scaffolds comprised a treatment with 10% gas phase fluorination in air for 60 s, a flushing process with synthetic air and additionally, an infiltration with a bovine collagen foam (soluble bovine acidic collagen was refibrillated with phosphate buffer and NaCl). The resulting hydrogel within and around the scaffold was desalted and lyophilized to form a foam between the threads of the embroidered polymer and afterwards the collagen was stabilized by cross-linking which was performed with hexamethylene diisocyanate (HMDI, Merck, Darmstadt, Germany) in an exiccator. This was done at the FILK (Freiberg, Germany) and already described [15,19]. To sum up, the functionalization (= modification) of embroidered scaffolds comprised gas phase fluorination and the integration of a HMDI cross-linked collagen foam.

### 4.3. Isolation of Fibroblasts from Lapine ACLs

Fibroblasts from the LACL were isolated from ACLs of 11 healthy (8 female and 3 male) New Zealand rabbits (approximate 14 months old) derived from the abattoir. Explanted LACLs were sliced into 2 mm^2^ pieces and placed in a T25 culture flask (Sarstedt AG & Co. KG, Nürnbrecht, Germany) with growth medium (Dulbecco’s Modified Eagle’s Medium (DMEM)/Ham’s F12 medium (Bio&SELL GmbH, Feucht, Germany) supplemented with 10% fetal bovine serum (FBS, Bio&SELL GmbH, Feucht, Germany), 1% penicillin/streptomycin solution (Bio&SELL GmbH, Feucht, Germany), 25 µg/mL ascorbic acid (Sigma-Aldrich, Munich, Germany), 2.5 µg/mL amphotericin B (Bio&SELL GmbH, Feucht, Germany) and MEM amino acid solution (Bio&SELL GmbH, Feucht, Germany) for several weeks. Growth medium changes were performed every second to third day. Emigrating cells were harvested with 0.05% trypsin/0.02% ethylenediaminetetraacetic acid (EDTA, Carl Roth GmbH, Karlsruhe, Germany) solution after 7–10 days and could be further expanded up to passage 5.

As a control, 3 native LACLs (mean length of 0.68 ± 0.19 cm) were used in the CyQuant and DMMB Assays (see the description below) and also for the real time PCR analysis (see the description below).

### 4.4. Scaffold Colonization

Non-functionalized (control) and functionalized scaffolds were sterilized in 70% ethanol (EtOH, Carl Roth GmbH, Karlsruhe, Germany) for a minimum of 30 min. After three times washing with low pyrogen content, sterile and hypotonic water (Carl Roth GmbH, Karlsruhe, Germany), the scaffolds were pre-incubated in FBS for 30 min. Sterilized scaffolds were then colonized with two different strategies.

### 4.5. Cell Suspension-Based Scaffold Colonization

For the cell suspension-based seeding strategy (“suspension pre-culture”), the scaffolds were transferred into TubeSpin bioreactor tubes (TPP, Trasadingen, Switzerland) and colonized with a cell suspension consisting of 8333.3 LACL-derived fibroblasts/mm^3^ suspended in growth medium. The cultivation was performed in 10 mL growth medium for one day on a rotatory device (Bartelt GmbH, Graz, Austria) with 36 rounds per minutes (rpm) at 37 °C. During this time no medium change was done before the scaffolds were transferred to the 3D stretch device and the 3D stretch protocol started. For the unstimulated controls, the scaffolds were after the one-day pre-cultivation transferred into the frame and put into a sterile beaker with 30 mL cultivation medium.

### 4.6. Spheroid-Based Scaffold Colonization

Spheroids were achieved by using the hanging drop method as described previously [15] with a cell number of 5.0 × 10^4^ per spheroid. After two days of spheroid formation at 37 °C and 5% CO_2_, they were carefully harvested. Scaffolds were colonized with 20 spheroids to guarantee the same cell number as in the suspension-based scaffold colonization. The spheroid colonized scaffolds (“spheroid pre-culture”) were statically incubated in growth medium for 5 days at 37 °C and 5% CO_2_ to allow the spheroid adhesion before they were transferred to the 3D stretch device for 3 days. For the unstimulated controls, the scaffolds were transferred after the five-day pre-cultivation into the frame and put into a sterile beaker with 30 mL cultivation medium.

### 4.7. Three-Dimensional (3D) Mechanostimulator and Cyclic Stretching

The custom-made 3D mechanostimulator consists of a chamber and a stretching device (Figure 1H) which can be controlled by a self-programmed software. The chamber has a filling volume of about 30 mL. The scaffold to be stretched is clamped in a stretch frame. The upper side of the scaffold is fixed with a stretch clamp attached to a pull rod. By fixing the pull rod to the stretch lever, an even stretching can be applied to the scaffold. Stretch of 4% at 0.11 Hz were performed, with a speed of 2.4 mm per second (mm/s) and a dwell time of 4.2 s. The release term was reached with a speed of 2.4 mm/s and a dwell time of 4.2 s. The amplitude was 0.6 mm. One cycle had a duration of 8.9 s. The duration of the entire cyclic stretching period was 3 days.

### 4.8. Cell Survival

For vitality assay a stain solution was mixed, containing 1 mL PBS, 1 µL propidium iodide (PI, 1% stock solution, Thermo Fisher Scientific, Darmstadt, Germany) and 5 µL fluorescein diacetate (FDA, stock solution: 3 mg/mL in acetone, Sigma-Aldrich, Munich, Germany) to visualize the live and dead fibroblasts on the scaffold after the cyclic stretch process. Scaffolds were longitudinally halved and after removing the medium from the scaffold, 50 µL of stain solution was added and transferred to a microscopic cover slide. After a 5-min incubation period at room temperature (RT), the fluorescence of live and dead cells was monitored using a Leica TC SPII confocal laser scanning microscope (CLSM, Leica Microsystems GmbH, Wetzlar, Germany). Based on the pictures with only vital cells, the colonized area was measured with the ImageJ 1.48v software (National Institutes of Health, MD, USA). Three independent experiments were performed with three different observation microscopic fields by the CLSM. In summary, nine microscopic fields on each scaffold were analyzed and the measured values were related to the whole scaffold size.

### 4.9. Immunocytochemical Staining and Measurement of the Migration Depth of Cells into the Scaffold

The protein expression profile was assessed using CLSM. Suspension- and spheroid cultured scaffolds (each *n* = 3) were fixed in 4% paraformaldehyde (PFA, Sigma Aldrich, St. Louis, MO, USA) for 30 min, washed with Tris buffered saline (TBS: 0.05 Tris, 140 mM NaCl, pH 7.6, Carl Roth GmbH, Karlsruhe, Germany), before incubation with blocking buffer (5% protease free donkey serum diluted in TBS with 0.1% Triton X 100 for cell permeabilization) was performed for 20 min at room temperature (RT). Samples (for each experiment a third of the scaffold) were incubated with primary antibodies (see Table 1: mouse anti human-αSMA and Ki67) over night at 4 °C. After rinsing with TBS, samples were incubated for 1 h with donkey-anti-mouse-cyanine-3-(Cy3, Invitrogen, Waltham, MA, USA) coupled secondary antibodies (diluted 1:200 in blocking buffer, see Table 1) at RT. The cell nuclei were counterstained using 10 μg/mL 4′,6′-diamidino-2-phenylindol (DAPI, Roche, Mannheim, Germany). Phalloidin-Alexa-Fluor 488 (1:100, Santa Cruz Biotechnologies, Inc., Dallas, TX, USA) was used to depict the filamentous (F)-actin cytoskeletal architecture. After three times of washing with TBS, samples were examined by CLSM.

### 4.10. Histological Staining of Sirius Red

The scaffolds were stained as a whole. Then, they were subjected to a descending EtOH series (96%, 80%, 70% and 60%) (all EtOH were obtained from Carl Roth GmbH, Karlsruhe, Germany), 4 min each step before washing in aqua nondest. for 4 min. Then, the cell nuclei were stained with Weigerts hematoxylin (MORPHISTO GmbH, Frankfurt am Main, Germany) for 8 min and the samples were washed with aqua dest. for 5 s, rinsed in aqua nondest. for 10 min before washed again in aqua dest. for 1 min. The staining with the Sirius Red solution (MORPHISTO GmbH, Frankfurt am Main, Germany) was performed for 60 min. Two steps in 30% acetic acid (Carl Roth GmbH, Karlsruhe, Germany) for 1 min each and followed by two steps in 96% EtOH for 4 min each. The scaffolds were stored in isopropanol (Carl Roth GmbH, Karlsruhe, Germany) and photographed with a light microscope (Diaplan, Leica Microsystems GmbH, Wetzlar, Germany).

### 4.11. Quantitative Assays for DNA- and sGAG Quantification

The quantitative DNA amount of colonized scaffolds and the native LACL was measured with the CyQuant^®^NF Cell proliferation Assay (Thermo Fisher Scientific Inc., Darmstadt, Germany). The standard curve was generated by serial dilution of calf thymus DNA stock solution (1 mg/mL) with TRIS/EDTA (TE)-buffer (10 mM TRIS (pH 8.0), 1 mM EDTA in Aqua dest.). For the standard curve, 25 µL of the serial calf thymus DNA dilutions was mixed with 25 µL of CyQuant dye solution (Hank‘s balanced salt solution (HBSS, Carl Roth GmbH, Karlsruhe, Germany) + dye binding solution 1:250 (Thermo Fisher Scientific Inc., Darmstadt, Germany)). After the stretch, scaffolds were cut small and had to be transferred to RNase and DNase using free 2 mL safe seal tubes (Sarstedt AG & Co. KG, Nürnbrecht, Germany) with 50 µL of the Proteinase K digestion buffer (50 mM Tris/HCl, 1 mM EDTA, 0.5% Tween 20) containing 0.5 mg/mL proteinase K (PanReac, ApplyChem, Darmstadt, Germany). Samples were homogenized with a 7 mm stainless steel bead (RNase and DNase free, sterile, Qiagen, Hilden, Germany) by using tissue lyser (Qiagen, Hilden, Germany, 50 Hz, 5 min, RT). Then, 250 µL of the Proteinase K digestion buffer (50 mM Tris/HCl, 1 mM EDTA, 0.5% Tween 20) containing 0.5 mg/mL proteinase K (PanReac, ApplyChem, Darmstadt, Germany) were added. All samples were digested for 16 h at 56 °C under continuous shaking (36 rpm). The enzymatic reaction was stopped by freezing the samples at −20 °C for 30 min. Before DNA quantification, all samples were thawed and then centrifuged for 15 min at 10,000 rpm. 10 µL of each sample were added to 150 µL TE buffer and thoroughly mixed. Samples were transferred in triplicate with 25 µL of the sample dilution to a black, flat bottom 96-well plate (Brand GmbH, Wertheim, Germany) and mixed with 25 µL of the dye solution (HBSS + dye solution 1:250). Subsequently, plates were covered to be protected from light and incubated at 37 °C for 60 min. The fluorescence of each well was measured in triplicate at 485 excitation/530 emission nm in a fluorometric plate reader (Tecan, Groedig, Austria). Three independent experiments with cells derived from three different donors were performed.

For the dimethyl methylene Blue (DMMB) Assay, the same supernatant was used as in the CyQuant Assay. After adequate sample dilution the DMMB solution (ApplyChem Darmstadt, Germany) was added (40 mM glycine (Sigma-Aldrich, Munich, Germany), 40 mM NaCl (Carl Roth GmbH, Karlsruhe, Germany) at pH 3 and DMMB (8.9 mM in ethanol)). Chondroitin sulfate (Sigma-Aldrich, Munich, Germany) was used as standard. The absorption shift was measured at wave length of λ = 633 nm to λ = 552 nm using a Genios spectral photometer (Tecan, Groedig, Austria). Three independent experiments with cells derived from three different donors were performed.

### 4.12. RNA Isolation

Native LACLs and stretched and non-stretched colonized scaffolds were snap-frozen (each *n* = 3). Samples immersed in 300 µL in RLT-buffer (Qiagen, Hilden, Germany) were homogenized 2 times with a tissue lyser (Qiagen, Hilden, Germany) each run at 50 Hz for 3 min. RNA was isolated and purified using the RNeasy Mini kit according to the manufacturer’s instructions (Qiagen, Hilden, Germany), including on-column DNase treatment. Purity (calculated from 260/280 absorbance ratio) and quantity of the RNA samples were monitored using the Nanodrop ND-1000 spectrophotometer (Peqlab, Biotechnologie GmbH, Erlangen, Germany).

### 4.13. Quantitative Real-Time PCR

The cDNA synthesis was carried out with 120 ng of total RNA and the reverse transcription kit (QuantiTect Reverse Transcription Kit, Qiagen, Hilden, Germany) according to the supplier manual. For each quantitative real-time PCR (qRT-PCR) reaction a total of 20 ng cDNA were used of TaqMan Gene Expression Assay (Life Technologies, Darmstadt, Germany) with primer pairs for type I collagen (*COL1A1*), decorin (*DCN*), tenascin C (*TNC*), tenomodulin (*TNMD* synonymous: myodulin), Mohawk (*MKX*) and connexin 43 (*CXN43*) and the housekeeping gene glyceraldehyde 3-phosphate dehydrogenase (*GAPDH*) as a reference gene (Table 2). The real time PCR detector StepOnePlus (Applied Bioscience (ABI), Foster City, CA, USA) thermocycler with the program StepOnePlus software 2.3 (ABI, Foster City, CA, USA) was used for qRT-PCR. The relative expression of the gene of interest by the cells on the scaffolds was normalized to the *GAPDH* expression and calculated for each sample using the ∆CT method as described [91].

### 4.14. Image Analyses

Based on the vitality images, the statistical analysis of colonized area on the scaffold was calculated with the ImageJ1.48v software. The pictures were split in the “green” and “red” channels and after converting into a “binary picture”, the area of the dots was measured and calculated as the colonized scaffold surface.

The migration depth of the cells into inner parts of the scaffolds was calculated with DAPI stained scaffolds. The degree of cell penetration of vertically cross-sectioned scaffolds was measured with the Leica X 3D image simulation program (Leica Microsystems GmbH, Wetzlar, Germany). Measurement was done starting at the scaffold border extending to the position of the innermost nuclei detected in the scaffold. The absolute values were taken for the statistic. Three independent experiments were performed. In summary, fifteen images were evaluated of each scaffold variant (five different positions per scaffold were taken to evaluate the migration depth). The proliferation rate (Ki67) and the percentage of αSMA per cell were calculated by splitting the pictures (in total nine pictures, three technical and three biological replicas) in the blue, green and red channels and counting the amount of blue dots (cell nuclei) and the amount of red dots (cell nuclei of proliferating cells) or the red area (αSMA).

### 4.15. Statistics

All values (colonized area, migration depth, proliferation rate, αSMA per cell, CyQuant Assay and DMMB Assay) were expressed as the mean with the standard deviation (SD) and relative gene expression were expressed as normalized to the native LACL values with the standard error of mean (SEM) using GraphPad Prism 8 (GraphPad Software Inc., San Diego, CA, USA). Before testing the normal distribution of the results, the ROUT method of identifying outliers was applied. The normal distribution of the results was analyzed using one-way ANOVA. Multiple comparison (Tukey post hoc test) was performed to evaluate the significance between the groups. Statistical significance was set at a *p*-value ≤ 0.05 (*), *p*-value ≤ 0.01 (**), *p*-value ≤ 0.0005 (***) and *p*-value ≤ 0.0001 (****). Three independent experiments with cells derived from three different donors were performed.

## 5. Conclusions

A functionalization of the scaffolds with 10% gas phase fluorination and a HMDI cross-linked collagen foam led to a superior outcome in regard to LACL cell colonization, sGAG content, and their relative gene expression of type I collagen in comparison to the control scaffolds.

Mechanostimulation of embroidered (P(LA-CL))/PLA scaffolds resulted in a more uniform colonization by LACL derived fibroblasts regardless of the pre-cultivation strategy applied when compared to unstimulated controls. Furthermore, stretch could mediate cell proliferation and myofibroblast transition. Nevertheless, looking exclusively on the pre-culturing strategy selected, the relative gene expression of type I collagen, tenascin C, tenomodulin and Mohawk indicated that a pre-cultivation of unstimulated scaffolds with spheroids stabilized ligamentocyte phenotype compared to the suspension culture and might allow a directed seeding of particular scaffold areas with cells. One limitation of the study is that protein expression was only demonstrated for sGAGs, total collagen, Ki67 and αSMA but not for the ligament-related proteins demonstrated at the gene expression level.

## Figures and Tables

**Figure 1 ijms-22-08204-f001:**
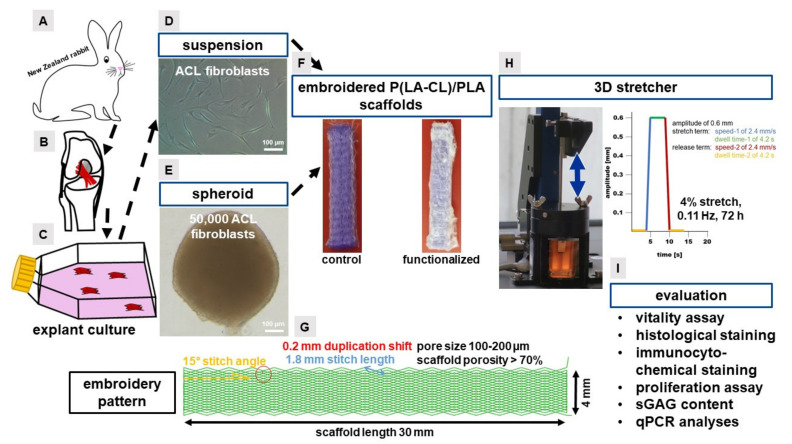
Scheme of the experimental setting. New Zealand rabbits (**A**) are the tissue and cell donors for this study. The anterior cruciate ligament ((**B**), marked in red) was dissected from the knee joint and cultured via explant culture (**C**). Two cultivation strategies, the suspension (**D**) and the spheroid culture (**E**) were performed for scaffold colonization. Non-functionalized (control) and functionalized embroidered poly(l-lactide-co-ε-caprolactone (P(LA-CL)) scaffolds (30 mm × 4 mm × 1 mm) were compared (**F**). The scaffold embroidery (**G**) consisted of a zigzag pattern with 1.8 mm stitch length, 15° stitch angle and 0.2 mm duplication shift. After pre-cultivating the scaffolds with cells, scaffolds were put into a 3D stretcher and 4% stretched (0.11 Hz) for 72 h (**H**). The cycle starts with a pause time (=dwell time-2 of 4.2 s, yellow line). In this phase the scaffolds were in a “zero position”. Then, the scaffolds were stretched with a speed of 2.4 mm/s (blue line) to reach the defined stretch position of 4% (0.6 mm amplitude). The stretched scaffold remained in this position for 4.2 s (=dwell time-1, green line). Then, the scaffolds were transferred to the starting position with a speed of 2.4 mm/s (red line) and left in this position for 4.2 s (=dwell time-2, yellow line). This stretch cycle was continuously repeated for 72 h (**H**). Evaluation (**I**) included vitality assay, histological staining, immunocytochemical staining, DNA and sulphated glycosaminoglycan (sGAG) content measurements and real time polymerase chain reactions (qPCR).

**Figure 2 ijms-22-08204-f002:**
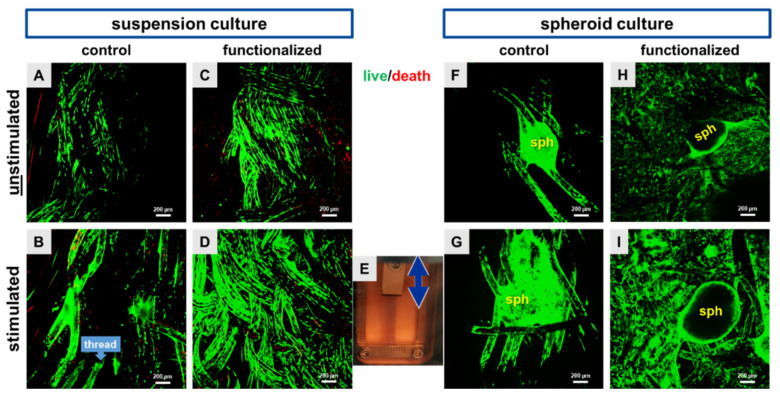
Representative scaffold pictures after the vitality assay, where live cells were depicted in green and dead cells were depicted in red. Both pre-cultivation strategies, the dynamical cell suspension (24 h) and the statical spheroid colonization (5 days) were performed on control scaffolds (**A**,**B**,**F**,**G**) and on functionalized scaffolds (**C**,**D**,**H**,**I**). Unstimulated (**A**,**C**,**F**,**H**) scaffolds were compared with stimulated (72 h, 4% stretch, 0.11 Hz, (**B**,**D**,**G**,**I**)) ones. A scaffold inside the stretch device (**E**) is shown, the double-headed arrow marks the stretch direction. Scale bars: 200 µm. Three independent experiments with cells of three different donors were performed.

**Figure 3 ijms-22-08204-f003:**
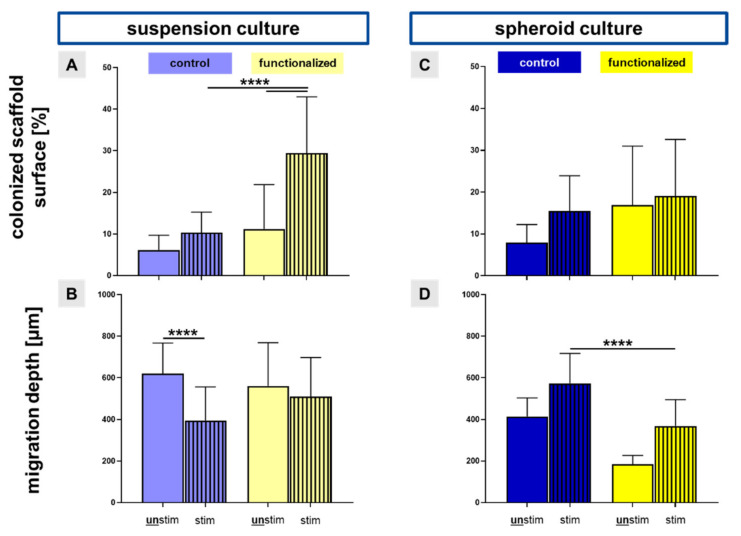
The statistic evaluation of the colonized scaffold surface and the migration depth into the scaffolds seeded with the suspension and the spheroid culture. Based on the vitality assay, the colonized scaffold surface ((**A**) **suspension** and (**C**) **spheroid pre-culture**) was evaluated. The migration depth of the cells into inner parts of the scaffolds ((**B**) **suspension** and (**D**) **spheroid pre-culture**) was measured after a DAPI staining of longitudinal sections. Control (non-functionalized, blue bars) and functionalized (yellow bars) scaffolds, seeded with two different colonization strategies (suspension: dynamically pre-cultured for 24 h and spheroid: statically pre-cultured for 5 days) were compared as well as unstimulated (unstim: full bars) against stimulated (72 h, 4% stretch, 0.11 Hz, stim: striped bars). One-way ANOVA test with Tukey post hoc test showed significant differences (****) with *p* ≤ 0.0001. Three independent experiments with cells of three different donors were performed.

**Figure 4 ijms-22-08204-f004:**
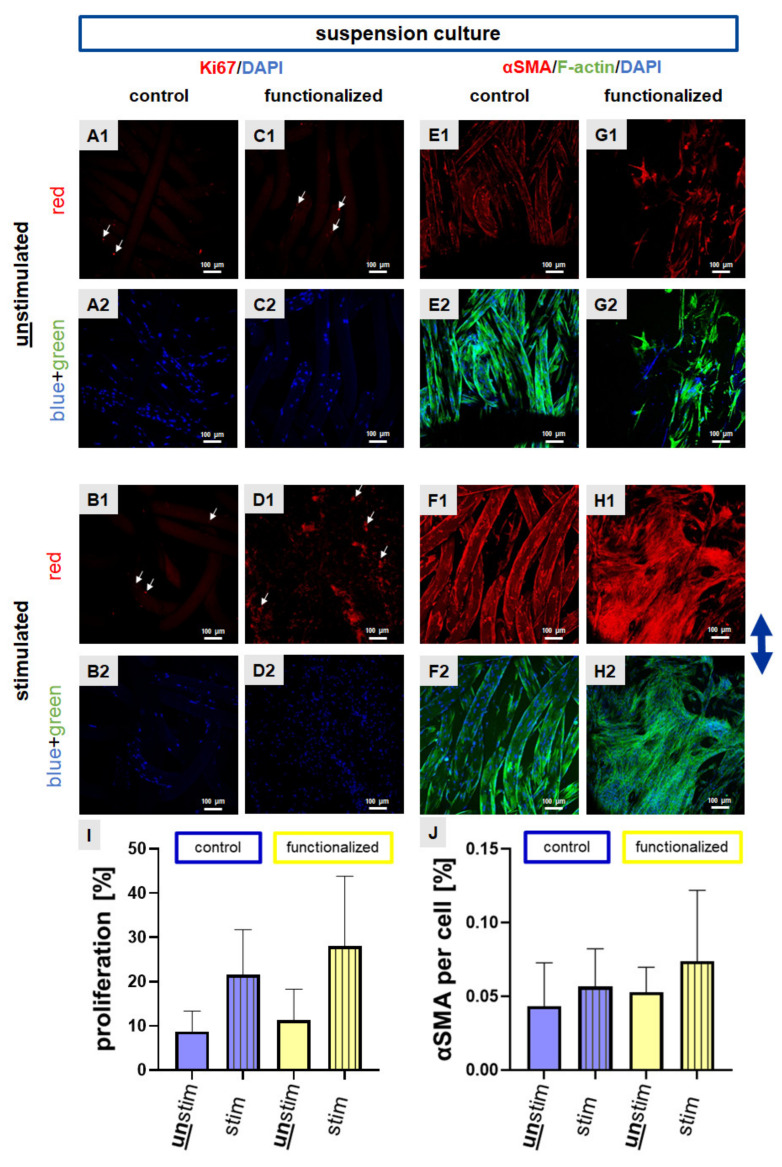
Ki67 and αSMA protein expression in scaffolds seeded with the suspension culture. Suspension pre-culture was performed for 24 h dynamically. Immunocytochemical staining of Ki67 (red) was performed for control (**A**,**B**) and functionalized scaffolds (**C**,**D**). Immunocytochemical staining of the myofibroblast marker α-smooth muscle actin (αSMA, red) combined with F-actin staining (green) was performed for control (**E**,**F**) and functionalized scaffolds (**G**,**H**). Unstimulated (**A**,**C**,**E**,**G**) scaffolds were compared with stimulated (72 h, 4% stretch, 0.11 Hz, (**B**,**D**,**F**,**H**)). The proliferation rate (**I**) and αSMA immunoreactivity per cell (%) (**J**) were calculated. Cell nuclei were counterstained using 4′,6′-diamidino-2-phenylindol (DAPI, blue). The blue arrow on the right side indicates the stretch direction. For the overlay of immunocytochemical staining please refer to Appendix A. Scale bars: 100 µm. Three independent experiments with cells of three different donors were performed. Unstim = unstimulated, stim = stimulated. Mean values with SD were shown in the statistical evaluation. One-way ANOVA testing did not show significant differences.

**Figure 5 ijms-22-08204-f005:**
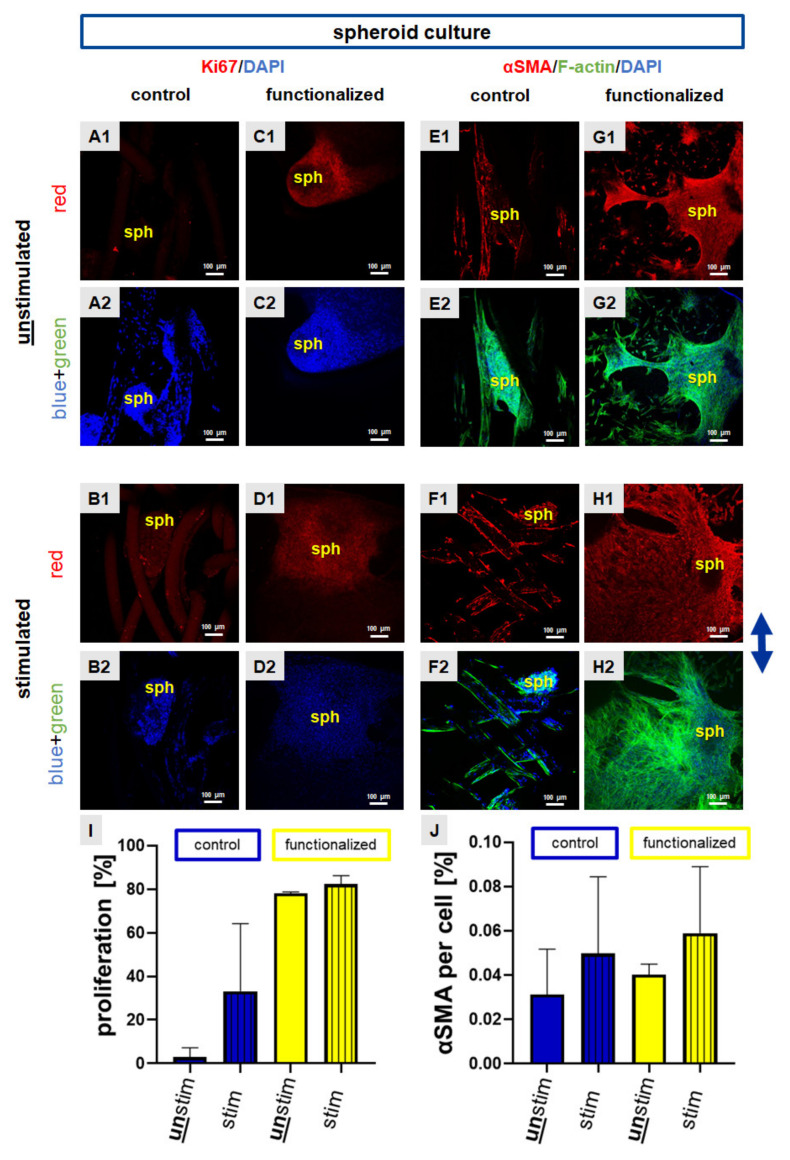
Ki67 and αSMA protein expression in scaffolds seeded with the spheroid (sph) culture. Scaffolds were statically pre-cultured for 5 days. Immunocytochemical staining of Ki67 (red) was performed for control (**A**,**B**) and functionalized scaffold (**C**,**D**). Immunocytochemical staining of the myofibroblast marker alpha smooth muscle actin (αSMA, red) combined with staining of F-actin (green) was performed for control (**E**,**G**) and functionalized scaffolds (**F**,**H**). Unstimulated (**A**,**C**,**E**,**G**) scaffolds were compared with stimulated (**B**,**D**,**F**,**H**). The proliferation rate (**I**) and αSMA immunoreactivity per cell (%) (**J**) were calculated. Cell nuclei were counterstained using 4′,6′-diamidino-2-phenylindol (DAPI, blue). The blue arrow on the right side indicates the stretch direction. For the overlay of immunocytochemical staining please refer to Appendix A. Scale bars: 100 µm. Three independent experiments with cells of three different donors were performed. Unstim = unstimulated, stim = stimulated. Mean values with SD were shown in the statistical evaluation. One-way ANOVA testing did not show significant differences.

**Figure 6 ijms-22-08204-f006:**
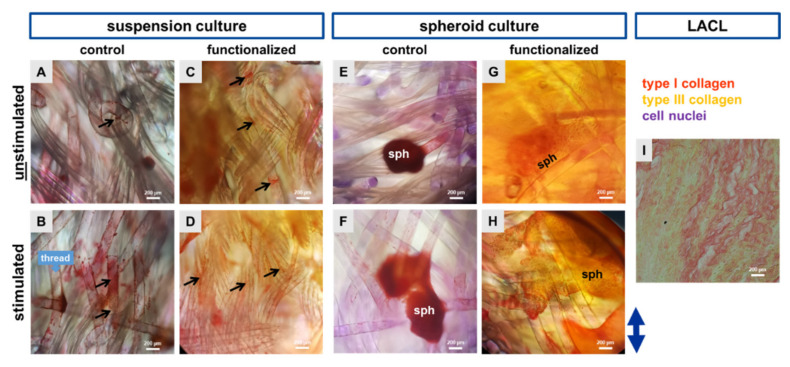
Representative pictures presenting an overview of the colonized scaffolds stained by Sirius Red. Control scaffolds seeded with cell suspension (dynamically pre-cultured for 24 h) either unstimulated (**A**) or stimulated (**B**) and functionalized scaffolds either unstimulated (**C**) or stimulated (**D**). Spheroid (sph) pre-colonized control scaffolds (statical pre-culture for 5 days) either unstimulated (**E**) or stimulated (**F**) and functionalized scaffolds either unstimulated (**G**) or stimulated (**H**). The double headed blue arrow (right side) indicated the stretch direction. The native lapine anterior cruciate ligament (LACL) is shown to see the thicker and mature type I collagen fibers (red-orange), the thinner type III collagen fibers (yellow) and the cell nuclei (violet) (**I**). Scale bars: 200 µm. The samples of one experiment were stained. Black arrows: pericellular type I collagen staining depicted in red.

**Figure 7 ijms-22-08204-f007:**
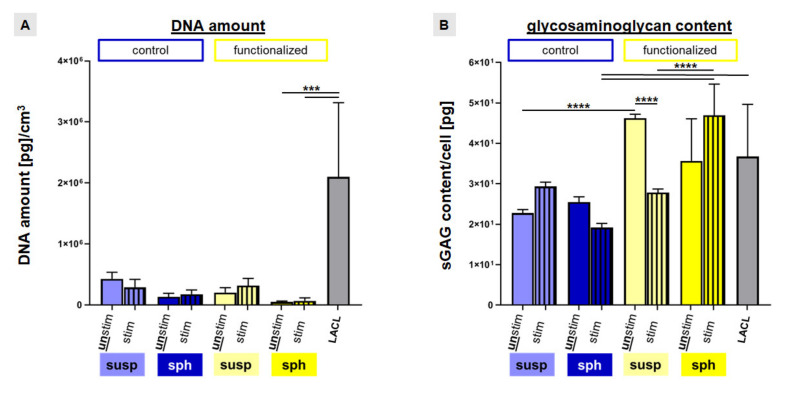
DNA amount and sulphated glycosaminoglycan (sGAG) content of colonized scaffolds in comparison to the native lapine anterior cruciate ligament (LACL). The DNA amount per cubic centimeter (**A**) and the sGAG content per cell (**B**) were evaluated in control scaffolds (blue) and functionalized scaffolds (yellow) with two different colonization strategies (suspension = susp: 24 h dynamically pre-cultured and spheroid = sph, 5 days statically pre-cultured). Additionally, the effect of stretch (striped bars) was evaluated. DNA amount and sGAG content of the native lapine ACL [LACL] were shown (grey). Three independent experiments with cells of three different donors were performed. One-way ANOVA test with Tukey post hoc test showed significant differences (***) with *p* ≤ 0.0005 and (****) with *p* ≤ 0.0001.

**Figure 8 ijms-22-08204-f008:**
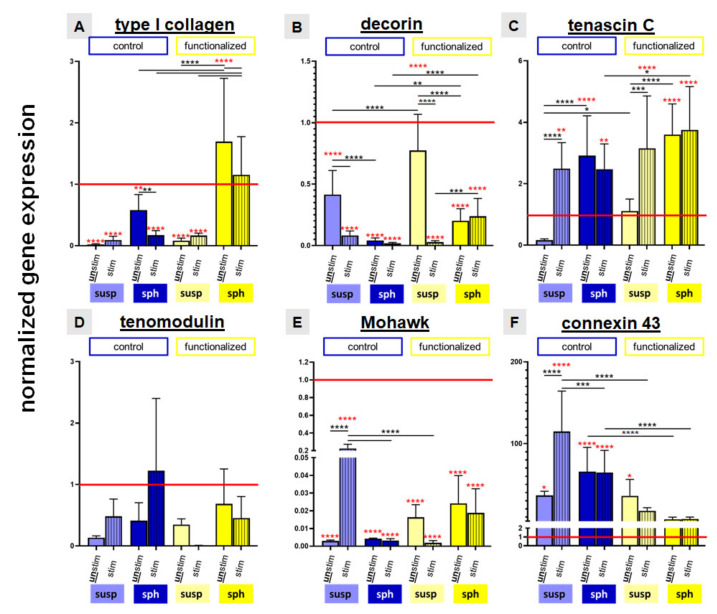
Normalized gene expression of ligament associated matrix components like type I collagen (**A**), decorin (**B**), tenascin C (**C**), the transcription factors tenomodulin (**D**) and Mohawk (**E**) and connexin 43 (**F**). Control scaffolds (blue) and functionalized scaffolds (yellow) were pre-colonized with suspension (susp: dynamically pre-cultured for 24 h) or spheroid (sph: statically pre-cultured for 5 days) culture. Unstimulated (unstim, full bars) and stimulated (stim, striped bars) scaffolds were compared to the native lapine anterior cruciate ligament (LACL, red line) and shown as mean with SEM. Three independent experiments with cells of three different donors were performed. One-way ANOVA test with Tukey post hoc test showed significant differences with (*) *p* ≤ 0.05; (**) *p* ≤ 0.01; (***) *p* ≤ 0.0005; (****) *p* ≤ 0.0001. Red stars indicated the significant differences in comparison to the native LACL.

**Table 1 ijms-22-08204-t001:** Antibodies used for immunocytochemistry.

Target	Primary Antibody	Dilution	Secondary Antibody	Dilution
Ki67	mouse-anti-human, Merck	1:30	donkey-anti-mouse Cy3, Invitrogen	1:200
αSMA	mouse-anti-human, Sigma Aldrich	1:50	donkey-anti-mouse Cy3, Invitrogen	1:200

Cy3: cyanine-3.

**Table 2 ijms-22-08204-t002:** Primer list for gene expression analysis.

Gene Symbol	Species	Gene Name	Amplicon Length	Assay ID
*COL1A1*	*O. cuniculus*	type I collagen, alpha 1	70	Oc03396073_g1
*CXN43*	*O. cuniculus*	connexin 43	68	Oc03396056_g1
*DCN*	*Homo sapiens*	decorin	77	Hs00370384_m1
*GAPDH* (LOC100009074)	*O. cuniculus*	glyceraldehyde-3-phosphate dehydrogenase	82	Oc03823402_g1
*MKX*	*O. cuniculus*	Mohawk	60	Oc06754037_m1
*TNC*	*O. cuniculus*	tenascin C	61	Oc06726696_m1
*TNMD* (LOC100125994)	*O. cuniculus*	myodulin	146	Oc03399505_m1

O.: Oryctolagus.

## Data Availability

Data supporting reported results can be provided by contacting the first author.

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
