# Peer review of "Maintenance of Ligament Homeostasis of Spheroid-Colonized Embroidered and Functionalized Scaffolds after 3D Stretch"

_ijms, 2021, doi:10.3390/ijms22158204_

Round 1
Reviewer 1 Report
This paper introduces an application of spheroids precultured on functionalized poly(L-lactide-co-ε-caprolactone) (P(LA-CL))/ polylactic acid (PLA) scaffolds followed by mechanical stretching. The evaluation of the colonized surface area, proliferation rate, αSMA content, and sGAG content of cells or some related gene expressions show the benefit of this strategy. However, some explanations should be revised for readers’ better understanding while some unclear points should be explained. Taken together, major revision should be made before paper re-submission.
- It’s better to insert the “Materials and Methods” before the “Results”, although this may be the style of the journal.
- 1. Cyclic stretch induced a homogenous cell distribution (Page 4, Figs2H, I)
Why is there no green fluorescence in the area inside spheroids?
- What about the size of spheroids? One of the current problems in 3D culture is that the poor permeability of oxygen and nutrition causes the death of cells After 5 days culture, how did the authors make sure that cells inside the spheroids still alive? It must be one of the reasons causes the low DNA amount (Fig 7A) of spheroid culture.
- Why is the spheroids culture better than the suspension culture? Related appropriate literatures or some results should be added to make the discussion about this point.
- 2. Migration into inner parts of the scaffold (Page 4, Line 143)
The figure should be Fig. 3C.
- 5. Enhanced cell amount and sGAG content by strain (Page 10, Lines252-254)
This sentence should be revised.
- Discussion (Page 13, Line 400)
It is claimed that after 72 hours in spheroid culture. However, it was different from the description in 4.6 (Page 17, Line 561).
- 1. Preparation of the embroidered P(LA-CL)/PLA scaffolds (Page 16)
What is the size of scaffolds? For readers’ better understanding, a schematic diagram or an image of scaffold structure should be added.
- Why were the PLA multifilaments used to prepare the scaffold? This point should be discussed comparing with other materials and formulation previously reported.
- 2. functionalization of the scaffolds (Page 16)
A detailed explanation of functionalization should be added.
The term “functionalized” is used in this study. What does it mean? What function is expected? More detailed explanations should be added.
- 7.3D Mechanostimulator and cyclic stretching (Page 17, Line 567)
The figure should be Fig. 1G.
- The stretching condition must greatly affect the functions of cell-scaffold constructs. This point should be discussed quoting some related literatures. The reason why the condition of 4% stretch at 0.18 Hz was selected in this study, should be explained.
- 8. Cell survival (Page 17, Line 582)
There is a thickness limitation in using confocal laser scanning microscope. How was the sample prepared for observation?
Author Response
Reviewer 1
This paper introduces an application of spheroids precultured on functionalized poly(L-lactide-co-ε-caprolactone) (P(LA-CL))/ polylactic acid (PLA) scaffolds followed by mechanical stretching. The evaluation of the colonized surface area, proliferation rate, αSMA content, and sGAG content of cells or some related gene expressions show the benefit of this strategy. However, some explanations should be revised for readers’ better understanding while some unclear points should be explained. Taken together, major revision should be made before paper re-submission.
- It’s better to insert the “Materials and Methods” before the “Results”, although this may be the style of the journal.
Response: we agree with the Reviewer, but this is the journal style. We are sorry for that.
- Cyclic stretch induced a homogenous cell distribution (Page 4, Figs2H, I)
Why is there no green fluorescence in the area inside spheroids?
Response: There is no green fluorescence inside the spheroids due to the fact, that our confocal laser scanning microscope (CLSM) has a defined maximum stack layer size of at least 500 µm (Griffith et al., 2006). By live-death cell imaging the cells were also alive inside the spheroids, but by focusing more on the cells emigrating from the spheroid onto the threads of the scaffold the upper part of the spheroid could not be photographed simultaneously. This is the reason why the spheroids look like “an egg cup”. It depends also on the condition how many threads stick out and thus, increase the distance between lens and spheroids. Since the core of this “cup” is also not stained red the presence of dead cells is not proven here.
- What about the size of spheroids? One of the current problems in 3D culture is that the poor permeability of oxygen and nutrition causes the death of cells. After 5 days culture, how did the authors make sure that cells inside the spheroids still alive? It must be one of the reasons causes the low DNA amount (Fig 7A) of spheroid culture.
Response: The spherical spheroids were photographed with the CLSM before scaffold precolonization and they had a mean diameter of 613.45 ± 175.96 µm. After scaffold colonization and after stretch the attached spheroids had a mean diameter of 593.57 ± 130.71 µm, a mean perimeter of 2277.67 ± 372.78 mm and a mean spheroid area of 366000 ± 114335.03 µm2 (Fig. 1A-B). This information was added now in section 2.1. It was not possible to measure the spheroids after 24 hours of colonization. Therefore, the size of the spheroids depends on their response to the scaffold surface and functionalization and the mechanical stimulus applied. Nevertheless, the reviewer is absolutely right with her/his comment, that the oxygen and nutrition permeability is limited. It can be expected that around 250 µm from each side of the spheroid can be supplied by diffusion. Hence, a hypoxic and poorly nourished core within the spheroid cannot be excluded. Nevertheless, during adhesion the cross sectional shape of the spheroids might become more flattened leading to a lower diffusion distance. This is supported by the 3D images shown below (Fig. 1C+D) and discussed in the second page of the discussion section (first paragraph). To achieve directed seeding of the scaffolds, spheroids of a size range large enough to prevent falling through scaffold pores between the threads were required (pore size of the scaffold between 100 – 200 µm). This important aspect was added to the discussion section now.
|
Figure 1A-D: Vitality staining of a representative spheroid (50.000 cells) before seeded on the scaffold after 2 d of spheroid assembly (A). B shows a spheroid on an embroidered scaffold after 8 days (including 3 days stretch). C and D depicts the 3D view of 8 day old spheroids on the scaffolds after mechanostimulation (C: control scaffold, D: scaffold functionalized) with a color code of their height. |
- Why is the spheroids culture better than the suspension culture? Related appropriate literatures or some results should be added to make the discussion about this point.
Response: the spheroid culture is superior due to the fact, that cells during expansion under 2D conditions tend to dedifferentiate (including a decrease of ligament related marker expression). Under the 3D conditions (=spheroids), the maintenance of ligament related matrix markers, more intense cell-cell interactions, and overall, stabilization of differentiated cell phenotype is an advantage (Barsby et al. 2014; Kraus et al. 2017; Wang et al. 2018). The 3D culture reproduces the native in vivo condition more closely than the 2D monolayer required for cell expansion and from which cells were directly detached to produce a cell suspension for scaffold seeding. The 3D spheroid culture is also very important to generate tumor organoid models (Schmidt et al. 2016; Sarkar et al. 2020) or to test drug inhibitory effects of particular agents (Berrouet et al. 2020).
The pre-culture of ligamentocytes in spheroids might allow redifferentiation of ligamentocytes expanded in 2D monolayer culture (Schulze-Tanzil et al., 2004; Stoll et al., 2010). In the present study spheroids bear the additional advantage of using them for directed seeding: they can directly be placed at distinct scaffold areas. In this study we found significantly higher sGAG content in response to stretching and type I collagen gene expression in functionalized scaffolds when using spheroid culture. This is stated at the end of the abstract. These advantages are already stated on page 2 of the introduction section.
- Migration into inner parts of the scaffold (Page 4, Line 143)
The figure should be Fig. 3C.
Response: the authors thank a lot the reviewer for this suggestion. The authors corrected and rewrote the whole phrase.
- Enhanced cell amount and sGAG content by strain (Page 10, Lines252-254)
This sentence should be revised.
Response: This and the incomplete sentence in the manuscript has been revised.
- Discussion (Page 13, Line 400)
It is claimed that after 72 hours in spheroid culture. However, it was different from the description in 4.6 (Page 17, Line 561).
Response: Yes, the time course of the steps of the spheroid culture was not very clearly written, we adapted it according to the description in the method section 4.6. We rewrote also the sentence: “Granato et al. (2017) showed that the expression of αSMA was downregulated after 72 hours in spheroid cultures, which indicated a myofibroblast deactivation.”
- Preparation of the embroidered P(LA-CL)/PLA scaffolds (Page 16)
What is the size of scaffolds? For readers’ better understanding, a schematic diagram or an image of scaffold structure should be added.
Response: The authors thanks for this remark and rephrased the paragraph describing the scaffold manufacturing. We added: “The scaffold had a dimension of 30 mm length, a width of 4 mm and a thickness of 2 – 3 mm. The pores were between 100 – 200 µm and the whole scaffold had a porosity of more than 70 %).” The picture below shows the embroidery pattern with the scaffold dimensions. It has been inserted now in Figure 1 of the manuscript.
Figure 2: The embroidery pattern of the scaffold consisted of a zigzag pattern with 1.8 mm stitch length, 15° stitch angle and 0.2 mm duplication shift.
- Why were the PLA multifilaments used to prepare the scaffold? This point should be discussed comparing with other materials and formulation previously reported.
Response: PLA is compared to PCL more brittle and stiffer, thus a PLA-monofilament with a diameter required for embroidery would not meet the necessary flexibility. Also a structure embroidered from a multifilament shows a smoother haptic and buckling ability. Even the higher surface area is seen as an advantage.
- functionalization of the scaffolds (Page 16)
A detailed explanation of functionalization should be added.
The term “functionalized” is used in this study. What does it mean? What function is expected? More detailed explanations should be added.
Response: the authors added additional information concerning the gas fluorination technique in section 4.2. “Functionalized” means that the embroidered scaffolds were modified with a gas phase fluorination and a hexamethylene diisocyanate (HMDI) cross-linked collagen foam inside and around the polymer threads. With the functionalization an improved cell adherence on the scaffold can be expected. This hypothesis is supported by references in the manuscript (Schröpfer et al., 2020 and Gögele et al., 2020).
- 3D Mechanostimulator and cyclic stretching (Page 17, Line 567)
The figure should be Fig. 1 G.
Response: The authors thank the reviewer for indicating this. They changed it in 1 H now since a subpart (scheme of embroidery pattern) was added to Figure 1.
- The stretching condition must greatly affect the functions of cell-scaffold constructs. This point should be discussed quoting some related literatures. The reason why the condition of 4% stretch at 0.18 Hz was selected in this study, should be explained.
Response: The condition of 4% stretch was chosen since this is a physiological condition (rabbit ACL ruptures occur if over 20% stretch is applied according to Hefti et al. 1991). We recalculated the stretch parameters and found that the Hz had to be corrected to 0.11 Hz. This is corrected throughout the manuscript now.
- Cell survival (Page 17, Line 582)
There is a thickness limitation in using confocal laser scanning microscope. How was the sample prepared for observation?
Response: As mentioned above, the maximum stack size of our CLSM is 500 µm. The calculation was performed with the 5x objective lens (and 10x ocular), hence, 50x magnification.

Reviewer 2 Report
The manuscript by Gögele et al. described the culture of lapine ACL fibroblasts on functionalized fibrous scaffolds, subjecting to cyclic loading. The manuscript was very clear to read and the results were presented professionally. However, there are some minor issues on the experimental work. It is suggested that the authors address these issues in the revision.
1. Mechanical stimulation:
- It is suggested that the authors describe the cyclic tests more specifically since it was a custom-built unit and perhaps a custom method of test. In particular, please describe the “dwell” time and its relationship with the frequency of 0.18 Hz. Was the “dwell” time included in the frequency.
- The authors described that a strain of 4% was applied to the fibrous scaffolds. Usually, polymer fibers yield at around 2%. This means that the scaffolds were permanently deformed. How did the authors ensure that the 3-day cyclic stretching was actually applied to the scaffolds?
- Polymer fibers will exhibit stress relaxation more significantly in liquid due to the viscoelastic behaviors. How did the authors ensure that stress relaxation was not happening during the dwell time, assuming the previous comments of 4% strain was in the elastic region?
2. Stimulation and cell attachments: In Figure 3, the authors showed the percentage of colonized cells at the scaffold surface and the depth of the cell migration. However, this information did not include the total cell counts. How can the authors ensure that mechanical stimulation did not lose seeded cells?
3. Spheroids activities after culture: Did the authors test the activities of the cell spheroids after two days of the culture? This will give a baseline for the unstimulated and stimulated groups.
Author Response
Reviewer 2
The manuscript by Gögele et al. described the culture of lapine ACL fibroblasts on functionalized fibrous scaffolds, subjecting to cyclic loading. The manuscript was very clear to read and the results were presented professionally. However, there are some minor issues on the experimental work. It is suggested that the authors address these issues in the revision.
- Mechanical stimulation:
- It is suggested that the authors describe the cyclic tests more specifically since it was a custom-built unit and perhaps a custom method of test. In particular, please describe the “dwell” time and its relationship with the frequency of 0.18 Hz. Was the “dwell” time included in the frequency.
Response: The figure below shows a detailed schematic drawing of the stretch cycle and is inserted in figure 1 in the manuscript now. The dwell time was included in the frequency which consists of amplitude, pause and velocity. The duration of the cycle was 8.9 seconds. The Figure shown below depicting the stretch conditions is now integrated in figure 1 of the manuscript and discribed in section 4.7. We recalculated the stretch parameters and found that the Hz had to be corrected to 0.11 Hz. This is corrected throughout the manuscript now.
Figure 3: The cycle starts with a pause time (=dwell time-2 of 4.2 seconds, yellow line). In this phase the scaffolds were in a “zero position”, then the scaffolds were stretched with a speed of 2.4 mm/seconds (blue line) to reach the defined stretch position of 4% (0.6 mm amplitude). The stretched scaffold remained in this position for 4.2 seconds (=dwell time-1, green line). Then, the scaffolds were transferred in the starting position with a speed of 2.4 mm/seconds (red line) and left in this position for 4.2 seconds (=dwell time-2, yellow line). This stretch cycle was continuously repeated for 72 hours.
- The authors described that a strain of 4% was applied to the fibrous scaffolds. Usually, polymer fibers yield at around 2%. This means that the scaffolds were permanently deformed. How did the authors ensure that the 3-day cyclic stretching was actually applied to the scaffolds?
Response: The scaffolds were fabricated by embroidery technology causing a porous structure formed by aligned polymeric threads. These fabrics show a structure deformation up to 20%. During the embroidery process, the threads become entangled, so that there is a kind of thread material reserve. This material is initially used when applying a deformation before the threads are directly loaded. The applied strain lies in the so-called "toe region" of the stress-strain curve of the scaffolds and thus, in the elastic deformation range and below the yield point. There has no material deformation been implemented on the polymeric threads Hahn et al., 2015; 2019. This peculiarity of these embroidered polymer scaffolds is mentioned in the discussion section now.
Polymer fibers will exhibit stress relaxation more significantly in liquid due to the viscoelastic behaviors. How did the authors ensure that stress relaxation was not happening during the dwell time, assuming the previous comments of 4% strain was in the elastic region?
Response: As the reviewer correctly points out, polymers exhibit viscoelastic material behavior and show different degrees of stress relaxation depending on the test conditions, for example. We investigated the stress relaxation behavior of our embroidered scaffolds under hydrolytic degradation conditions in a previous study (Hahn et al., 2019). We could demonstrate a relaxation behavior comparable to native ACL tissue with a fast relaxation in the first 60 seconds followed by a moderate relaxation after 180 seconds.
- Stimulation and cell attachments: In Figure 3, the authors showed the percentage of colonized cells at the scaffold surface and the depth of the cell migration. However, this information did not include the total cell counts. How can the authors ensure that mechanical stimulation did not lose seeded cells?
Response: The reviewer is completely right; we had measured the colonized scaffold surface. In addition, the total cell counts were measured with the CyQuant assay showing the DNA amount after stretching. It is possible to recalculate the total cell number based on the DNA content of one cell. Since we did unfortunately not assess the DNA content immediately before stretching we cannot estimate the degree of cell loss in response to stretching and dependent on the scaffold variant. Hence, we cannot exclude it. This possibility of cell loss during stretching is mentioned in the discussion section (page 13).” Since the DNA content did not significantly differ between the scaffold variants, the question arises whether stretching might be associated with some cell loss which hides the effects of proliferation. To prove this, the DNA contents before and after stretching have to be compared.”
- Spheroids activities after culture: Did the authors test the activities of the cell spheroids after two days of the culture? This will give a baseline for the unstimulated and stimulated groups.
Response: Yes, the authors had performed the DNA (A) and sGAG (B) content measurements per spheroid before cultivation and assessed also in one experiment the relative gene expression of type I collagen and tenascin C (C) of spheroids. But we preferred to compare our data with the native ACL, since we aim to get a tissue engineered construct resembling as much as possible the ACL.
Figure 4: DNA (A, n=3), sGAG (B, n=3) contents and relative gene expression of collagen type I and tenascin C (C, n=1) in spheroids before scaffold seeding (2 d).
References cited in the point by point reply
Barsby T, Bavin EP, Guest DJ (2014). Three-Dimensional Culture and Transforming Growth Factor β3 Synergistically Promote Tenogenic Differentiation of Equine Embryo-Derived Stem Cell. TISSUE ENGINEERING: Part A Volume 20, Numbers 19 and 20, 2604-2613.
Berrouet C, Dorilas N, Rejniak KA, Tuncer N (2020). Comparison of drug inhibitory effects (IC50) in monolayer and spheroid cultures. bioRxiv preprint doi: https://doi.org/10.1101/2020.05.05.079285.
Goegele C, Hahn J, Elschner C, Breier A, Schroepfer M, Prade I, Meyer M, Schulze-Tanzil G. Enhanced Growth of Lapine Anterior Cruciate Ligament-Derived Fibroblasts on Scaffolds Embroidered from Poly(l-lactide-co-epsilon-caprolactone) and Polylactic Acid Threads Functionalized by Fluorination and Hexamethylene Diisocyanate Cross-Linked Collagen Foams. Int J Mol Sci 2020, 21, doi:10.3390/ijms21031132.
Griffith LG, Swartz MA. Capturing complex 3D tissue physiology in vitro. Nat Rev Mol Cell Biol 2006, 7, 211-224, doi:10.1038/nrm1858.
Hahn J, Hinüber C, Breier A, Heinrich G (2015). Adjusting the mechanical behavior of embroidered scaffolds to lapin anterior cruciate ligaments by varying the thread materials. Textile Research Journal 85(14):1431-1444
Hahn J, Schulze-Tanzil G, Schröpfer M, Meyer M, Gögele C, Hoyer M, Spickenheuer A, Heinrich G, Breier A (2019). Viscoelastic Behavior of Embroidered Scaffolds for ACL Tissue Engineering Made of PLA and P(LA-CL) After In Vitro Degradation. Int. J. Mol. Sci. 2019, 20, 4655; doi:10.3390/ijms20184655.
Hefti F, Kress A, Fasel J, Morscher EW (1991). Healing of the Transected Anterior Cruciate Ligament in the Rabbit. J Bone Joint Surg VOL. 73-A, NO. 3, 373-382.
Kraus A, Luetzenberg R, Abuagela N, Hollenberg S, Infanger M (2017). Spheroid formation and modulation of tenocyte-specific gene expression under simulated microgravity. Muscles, Ligaments and Tendons Journal 2017;7 (3):411-417.
Sarkar S, Peng C-C, Tung Y-C (2020). Comparison of VEGF-A secretion from tumor cells under cellular stresses in conventional monolayer culture and microfluidic three-dimensional spheroid model. PLOS ONE | https://doi.org/10.1371/journal.pone.0240833 November 11, 202.
Schmidt M, Scholz C-J, Polednik C, Roller J (2016); Spheroid-based 3-dimensional culture models: Gene expression and functionality in head and neck cancer. ONCOLOGY REPORTS 35: 2431-2440. DOI: 10.3892/or.2016.4581.
Schroepfer M, Junghans F, Voigt D, Meyer M, Breier A, Schulze-Tanzil G,Prade I. Gas-Phase Fluorination on PLA Improves Cell Adhesion and Spreading. ACS Omega 2020, 5, 5498-5507, doi:10.1021/acsomega.0c00126.
Schulze-Tanzil G, Mobasheri A, Clegg PD, Sendzik J, John T, Shakibaei M (2004). Cultivation of human tenocytes in high-density culture. Histochem Cell Biol 122, 219-28.
Stoll C, John T, Endres M, Rosen C, Kaps C, Sittinger M, Ertel W, Kohl B, Schulze-Tanzil G (2010). Extracellular matrix expression of human tenocytes in three-dimensional air-liquid and PLGA cultures compared with tendon tissue: implications for tendon tissue engineering. J Orthop Res 28(9), 1170-7.
Wang T, Thien C, Wang C, Ni M, Gao J, Wang A, Jiang Q, Tuan RS, Zheng Q, Zheng MH (2018). 3D uniaxial mechanical stimulation induces tenogenic differentiation of tendon-derived stem cells through a PI3K/AKT signaling pathway. FASEB J. doi: 10.1096/fj.201701384R.

Reviewer 3 Report
General comment
The manuscript “Maintenance of ligament homeostasis of spheroid-colonized 2 embroidered and functionalized scaffolds after 3D stretch” by Gögele et al., is an interesting work in which Authors developed new 3D seeding strategy on 3D scaffold using spheroids to maintain ligamentogenesis in 3D compartment compared to the use of the conventional cell suspension seeding method.
Although the interesting in content and experimental design, the Authors did not adequately present the results that are confusing to the reader and do not fully support the conclusions.
The multiple panel figures are too much and the data does not exactly match with the images presented. Some critical points should be addressed in order to render the manuscript suitable for publication.
Major Critical points to be addressed:
-The Authors did use fluorescein diacetate to assess cell viability. Fluorescein acetate is intracellular, and it becomes fluorescent upon esterase hydrolysis. However, in Figure 2, the Authors present images to assess live/dead cells in which green fluorescence should show the live cells which should be localized intracellularly. Instead, in the presented figures, the green fluorescence seems to be an auto-fluorescence of the 3D scaffold since it is not only localized intracellularly. The Authors should to show Please clarify and discuss.
-The Figure 4 is very confused and figure legend should be better detailed reporting exactly the numbered images for investigation. Figure 4E did not reflect the Ki-67 level of cells showed in the relative images neither in the results in paragraph 2.3. The cells seem to not proliferate in the unstimulated culture conditions (A1 and C1) even though the histograms show relatively high proliferation rate especially for Control unstim. The Authors should address the results and change the figures showing images that are in accordance to the data Merge alpha-Sma/Dapi images are also required.
-The representative Images concerning the Ki-67 and a-SMA staining cells in both Figure 4 and 5, did not reflect the percentage of cell proliferation and a-SMA positive cells calculated and presented in the relative histograms.
Figure 5. A2 showed high number of migrated cells from the spheroids within the control scaffold cultured under unstimulated conditions in comparison to the already obtained results in the previous Figure 2G. Please discuss this discrepancy of results.
In the paragraph 2.5., the results of DNA and sGAG quantification lack the unit of concentration. Please fix.
In the line 254, there is a missing word. Please adjust.
Line 356, the sentence is not clear. The Authors indicated that the functionalized scaffolds contain collagen foam compared to the control scaffolds. Why functionalized scaffold might inhibit to a certain level cell penetration compared to control groups? Please discuss.
In line 379, the Authors talked about overload by applying 4% of stretch on cell seeded on scaffolds with no significant differences in cell proliferation, while when they used a 14% of stretch with the same cell type, they obtained higher proliferation rate. Please discuss and explain what overload means in this case.
Line 454, the Authors said “it is unclear why 4% stretching did not lead to an increase in the gene expression of type I collagen in the spheroid culture”, without specifying the reason or at least give a hypothesize concerning the obtained results since 4% represent physiological loading stretch in native tendon and ligament tissues. Please, discuss.
The Authors in Results refer to only gene expression of ligament related genes. There was no assessment of proteins expression to make firm conclusions and construct an understandable panel relative to the ligament related gene and protein expressions. The Authors to clarify and discuss these aspect and should indicate the study limitations in the Discussion section.
The Conclusions must be improved in accordance to Result and Discussion sections.
Author Response
Reviewer 3
Major Critical points to be addressed:
-The Authors did use fluorescein diacetate to assess cell viability. Fluorescein acetate is intracellular, and it becomes fluorescent upon esterase hydrolysis. However, in Figure 2, the Authors present images to assess live/dead cells in which green fluorescence should show the live cells which should be localized intracellularly. Instead, in the presented figures, the green fluorescence seems to be an auto-fluorescence of the 3D scaffold since it is not only localized intracellularly. The Authors should to show Please clarify and discuss.
Response: In this case it is not an autofluorescence: a rather low magnification was used for the images to present an overview over a large area of the colonized scaffold. Since areas with dense cells and those with few cells had to be simultaneously best possible depicted, the setting chosen for confocal microscopy was rather intense and hence, the green color of single cells covering the threads is fused. For this reason, the margins of the threads are marked by intense green representing the rim surrounding the threads consisting of cells. FDA can indead also lead to autofluorescence of the scaffolds. This is the case when the scaffold fibers start degradation due to prolonged culturing under wet conditions. Then, the surface becomes rough and the staining is captured. This leads to a more homogeneous background staining of the whole threads and not only the rims of them. In the present work, the culturing period on the scaffold was rather short (8 days for spheroid culture). In our previous publications (Gögele et al. 2020, especially Hahn et al. 2019 in Fig. 4 A1 and B1 the autoflourecence of the same scaffolds was shown, as well as in Schwarz et al. 2019). In the present work, we used FDA and no autofluorescence of the scaffolds in the green spectrum was detected. The green dye is only seen intracellularly. However, since there are a lot of cells wrapped around a thread, it looks like the thread is also "glowing".
-The Figure 4 is very confused and figure legend should be better detailed reporting exactly the numbered images for investigation. Figure 4E did not reflect the Ki-67 level of cells showed in the relative images neither in the results in paragraph 2.3. The cells seem to not proliferate in the unstimulated culture conditions (A1 and C1) even though the histograms show relatively high proliferation rate especially for Control unstim. The Authors should address the results and change the figures showing images that are in accordance to the data Merge alpha-Sma/Dapi images are also required.
Response: Relative proliferation refers to three independent experiments; the ki67 proliferation marker is only very faintly visible at this low magnification. The images were quantitatively analyzed with an image processing program, so that even low color intensities could be perceived and designated as "proliferation". But as written in the point below -> we have changed representative pictures and calculated the “proliferation rate” again.
The authors thank the reviewer for suggesting that an overlay of the images should be shown; We did it indeed in the beginning, but found that depicting the single channels allowed a clearer localization of the antigens hidden by the merged channels. Since the panels of images become to large to be shown in the manuscript, we can not show the overlays in addition within the same figure panel. Hence, we decided to show them as additional supplemental figure and we refer to it in the Legends of the respective figures of the manuscript.
-The representative Images concerning the Ki-67 and a-SMA staining cells in both Figure 4 and 5, did not reflect the percentage of cell proliferation and a-SMA positive cells calculated and presented in the relative histograms.
Response: The authors agree with the reviewer that this is not the total proliferation rate and αSMA expression. Only a representative part of the scaffold was stained, otherwise too much antibody would be consumed with a total scaffold length of 3 cm. Furthermore, we agree with the reviewer that total proliferation cannot be inferred exclusively based on protein expression of ki67. It is merely a snapshot in a selected scaffold region. We have changed the representative images in Fig 4 in a way that reflects the quantification more closely. We have also calculated the “proliferation rate” again and adapted the diagram accordingly.
Figure 5. A2 showed high number of migrated cells from the spheroids within the control scaffold cultured under unstimulated conditions in comparison to the already obtained results in the previous Figure 2G. Please discuss this discrepancy of results.
Response: There is a misunderstanding, because Fig. 5A2 shows the DAPI stain of an unstimulated spheroid located on the surface of a control scaffold.
However, the image Fig. 2G shows spheroids that were on a stimulated control scaffold. As already described in the discussion, a fusion of spheroids is often found during cultivation (as in Fig. 2G, so that the impression of a very large spheroid arises).
Furthermore, Fig. 2 and Fig. 5 can only be compared poorly with each other, because the images are taken at different magnifications. Also in Fig. 5, cells could be observed migrating out of the spheroid.
In the paragraph 2.5., the results of DNA and sGAG quantification lack the unit of concentration. Please fix.
Response: The authors thank the reviewer for the essential comment and have added the missing units to the manuscript.
In the line 254, there is a missing word. Please adjust.
Response: Thanks a lot, we have added the missing word in the sentence: In the absence of stretch functionalized scaffolds, seeded with a cell suspension displayed a significantly higher sGAG content compared to the unstimulated control scaffolds.
Line 356, the sentence is not clear. The Authors indicated that the functionalized scaffolds contain collagen foam compared to the control scaffolds. Why functionalized scaffold might inhibit to a certain level cell penetration compared to control groups? Please discuss.
Response: By this, the authors meant that by functionalizing with a collagen foam, cell migration into the interior of the scaffold is made more difficult. The additional collagen foam between the threads prevents the cells from penetrating deeper into the scaffold. Due to the manufacturing technique, the collagen foam may be rather dense and thus not all of its pores large enough for cell penetration. The reduced migration due to the foam is explained in more detail in the discussion section on page 13 now.
As mentioned earlier, the control scaffolds have no functionalization and therefore, no collagen foam, which means that cell migration is not disturbed.
In line 379, the Authors talked about overload by applying 4% of stretch on cell seeded on scaffolds with no significant differences in cell proliferation, while when they used a 14% of stretch with the same cell type, they obtained higher proliferation rate. Please discuss and explain what overload means in this case.
Response: The two culture systems (2D and 3D) can only be compared to a limited extent. While in 2D 14% stretch has a rather stimulating effect on the cells, in 3D already 4% can lead to some overload. The reason could possibly be the scaffold structure which three-dimensionally surrounds the cells. It could compress the cells when its pores become smaller during stretching and thus cells experience at the same % stretch a stronger load than in the 2D culture. In 2D cells can avoid stretching be changing their cell orientation in zero stretch direction. This cell orientation in an oblique or transverse direction in relation to the stretch direction is uncommon in 3D culture. Sheng et al. 2020 also underlined how complicated the comparability between 2D and 3D culture is. An additional reason could also be the “cultivation chamber” and therefore the medium amount and its content with soluble stimulatory factors (e.g. mechanogrowth factor and other growth factors) mediating cell-cell communication and other cell activities (e.g. 2 mL against 30 mL in the 3D system).
Line 454, the Authors said “it is unclear why 4% stretching did not lead to an increase in the gene expression of type I collagen in the spheroid culture”, without specifying the reason or at least give a hypothesize concerning the obtained results since 4% represent physiological loading stretch in native tendon and ligament tissues. Please, discuss.
Response: The authors hypothesize that the 4% from our 3D stretcher cannot directly be compared to the 4% of native tissue, as the spatial arrangement of cells in longitudinal rows in vivo as well as the unique native ACL ECM structure surrounding the cells is much denser and more complex. The cells seeded on the scaffolds still have a more or less random orientation. Furthermore, it is still unclear how much stretch actually reaches the individual cells, whether in a tissue engineered construct or in a tissue. Another hypothesis is that the stretch duration is too short to stimulate collagen type I synthesis in a 3D construct. During tendon regeneration (in stage III “rebuilding stage”) also collagen type III is upregulated very fast after an injury. The ratio between type I and III collagen has changed. Nevertheless, Wang et al. 2013 reported that there is a narrow range of tensile load promoting the tendon homeostasis. as Accordingly, Goncalves et al. 2018 mentioned “the optimal in vitro conditions have not been established”. In tissue engineering it might in addition to the stretch profile depend on many factors including cell number, growth medium, biomaterial, its biomechanics and topology. We discuss the accessiblility of serum in the spheroids triggered by fluid flow in response to stretching now in the discussion section as a hypothesis.
The Authors in Results refer to only gene expression of ligament related genes. There was no assessment of proteins expression to make firm conclusions and construct an understandable panel relative to the ligament related gene and protein expressions. The Authors to clarify and discuss these aspect and should indicate the study limitations in the Discussion section.
Response: The study is limited by the lack of analyzing ligament-associated protein expressions such as collagen type I, decorin, tenascin C and tenomodulin at the protein level. The panel of available antibodies reactive with rabbit tendon-related proteins is, unfortunately, rather limited. In previous experiments, attempts were made to show these matrix components, but so far we found no specific antibodies for this rabbit-derived cell type, for example for tenomodulin. Especially for collagen type I there were difficulties in immunohistology, because it was not possible to differentiate the freshly produced lapine collagen from the bovine collagen foam on the functionalized scaffolds. This was also one of the reasons why we decided against immunocytochemical staining of matrix components and showed collagen only with the Sirius red stain This clear limitation of our study has been integrated into the conclusion now.
The Conclusions must be improved in accordance to Result and Discussion sections.
Response: We reorganized and adapted the conclusion section accordingly. The main limitation of the study was integrated into the conclusion now.
References cited in the point by point reply
Barsby T, Bavin EP, Guest DJ (2014). Three-Dimensional Culture and Transforming Growth Factor β3 Synergistically Promote Tenogenic Differentiation of Equine Embryo-Derived Stem Cell. TISSUE ENGINEERING: Part A Volume 20, Numbers 19 and 20, 2604-2613.
Berrouet C, Dorilas N, Rejniak KA, Tuncer N (2020). Comparison of drug inhibitory effects (IC50) in monolayer and spheroid cultures. bioRxiv preprint doi: https://doi.org/10.1101/2020.05.05.079285.
Goegele C, Hahn J, Elschner C, Breier A, Schroepfer M, Prade I, Meyer M, Schulze-Tanzil G (2020). Enhanced Growth of Lapine Anterior Cruciate Ligament-Derived Fibroblasts on Scaffolds Embroidered from Poly(l-lactide-co-epsilon-caprolactone) and Polylactic Acid Threads Functionalized by Fluorination and Hexamethylene Diisocyanate Cross-Linked Collagen Foams. Int J Mol Sci 21, doi:10.3390/ijms21031132.
Gonçalves, A. I., Berdecka, D., Rodrigues, M. T., Reis, R. L., & Gomes, M. E. (2018). Bioreactors for tendon tissue engineering: challenging mechanical demands towards tendon regeneration. doi:10.1201/9780429453144
Griffith LG, Swartz MA. Capturing complex 3D tissue physiology in vitro. Nat Rev Mol Cell Biol 2006, 7, 211-224, doi:10.1038/nrm1858.
Hahn J, Hinüber C, Breier A, Heinrich G (2015). Adjusting the mechanical behavior of embroidered scaffolds to lapin anterior cruciate ligaments by varying the thread materials. Textile Research Journal 85(14):1431-1444
Hahn J, Schulze-Tanzil G, Schröpfer M, Meyer M, Gögele C, Hoyer M, Spickenheuer A, Heinrich G, Breier A (2019). Viscoelastic Behavior of Embroidered Scaffolds for ACL Tissue Engineering Made of PLA and P(LA-CL) After In Vitro Degradation. Int. J. Mol. Sci. 2019, 20, 4655; doi:10.3390/ijms20184655.
Hefti F, Kress A, Fasel J, Morscher EW (1991). Healing of the Transected Anterior Cruciate Ligament in the Rabbit. J Bone Joint Surg VOL. 73-A, NO. 3, 373-382.
Kraus A, Luetzenberg R, Abuagela N, Hollenberg S, Infanger M (2017). Spheroid formation and modulation of tenocyte-specific gene expression under simulated microgravity. Muscles, Ligaments and Tendons Journal 2017;7 (3):411-417.
Sarkar S, Peng C-C, Tung Y-C (2020). Comparison of VEGF-A secretion from tumor cells under cellular stresses in conventional monolayer culture and microfluidic three-dimensional spheroid model. PLOS ONE | https://doi.org/10.1371/journal.pone.0240833 November 11, 202.
Schmidt M, Scholz C-J, Polednik C, Roller J (2016); Spheroid-based 3-dimensional culture models: Gene expression and functionality in head and neck cancer. ONCOLOGY REPORTS 35: 2431-2440. DOI: 10.3892/or.2016.4581.
Schroepfer M, Junghans F, Voigt D, Meyer M, Breier A, Schulze-Tanzil G,Prade I. Gas-Phase Fluorination on PLA Improves Cell Adhesion and Spreading. ACS Omega 2020, 5, 5498-5507, doi:10.1021/acsomega.0c00126.
Schulze-Tanzil G, Mobasheri A, Clegg PD, Sendzik J, John T, Shakibaei M (2004). Cultivation of human tenocytes in high-density culture. Histochem Cell Biol 122, 219-28.
Schwarz S, Goegele C, Ondruschka B, Hammer N, Kohl B, Schulze-Tanzil G (2019). Migrating Myofibroblastic Iliotibial Band-Derived Fibroblasts Represent a Promising Cell Source for Ligament Reconstruction. Int J Mol Sci 2019, 20, doi:10.3390/ijms20081972.
Sheng R, Jiang Y, Backman LJ, Zhang W, Chen J. The Application of Mechanical Stimulations in Tendon Tissue Engineering. Stem Cells Int. 2020 Sep 24;2020:8824783. doi: 10.1155/2020/8824783. PMID: 33029149; PMCID: PMC7532391.
Stoll C, John T, Endres M, Rosen C, Kaps C, Sittinger M, Ertel W, Kohl B, Schulze-Tanzil G (2010). Extracellular matrix expression of human tenocytes in three-dimensional air-liquid and PLGA cultures compared with tendon tissue: implications for tendon tissue engineering. J Orthop Res 28(9), 1170-7.
Wang T, Lin Z, Day RE, Gardiner B, Landao-Bassonga E, Rubenson J, Kirk TB, Smith DW, Lloyd DG, Hardisty G, Wang A, Zheng Q, Zheng MH. Programmable mechanical stimulation influences tendon homeostasis in a bioreactor system. Biotechnol Bioeng. 2013 May;110(5):1495-507. doi: 10.1002/bit.24809. Epub 2013 Feb 4. PMID: 23242991.
Wang T, Thien C, Wang C, Ni M, Gao J, Wang A, Jiang Q, Tuan RS, Zheng Q, Zheng MH (2018). 3D uniaxial mechanical stimulation induces tenogenic differentiation of tendon-derived stem cells through a PI3K/AKT signaling pathway. FASEB J. doi: 10.1096/fj.201701384R.

Round 2
Reviewer 1 Report
All the comments are responded.
Now I recommend publishing the manuscript revised.
Reviewer 3 Report
The Revised Manuscript “Maintenance of ligament homeostasis of spheroid-colonized 2 embroidered and functionalized scaffolds after 3D stretch” by Gögele et al., resulted improved in all Sections and in Figures. Moreover, additional Supplementary file was added.
It is now appealing in content and in experimental design. The Authors fully satisfied the Reviewer's requests. Critical aspects have been clarified, results and figures have improved and support the Conclusions.